# BRIEF COMMUNICATION

## OPEN

# Structure of the decoy module of human glycoprotein 2 and uromodulin and its interaction with bacterial adhesin FimH

Alena Stsiapanava[1], Chenrui Xu[2,3], Shunsuke Nishio [1], Ling Han [1], Nao Yamakawa[4], Marta Carroni[5], Kathryn Tunyasuvunakool[6], John Jumper [6], Daniele de Sanctis [7], Bin Wu [2,3] and Luca Jovine [1,2 ✉]

**Glycoprotein 2 (GP2) and uromodulin (UMOD) filaments protect against gastrointestinal and urinary tract infections by acting as decoys for bacterial fimbrial lectin FimH. By combining AlphaFold2 predictions with X-ray crystallography and cryo-EM, we show that these proteins contain a bipartite decoy module whose new fold presents the high-mannose glycan recognized by FimH. The structure rationalizes UMOD mutations associated with kidney diseases and visualizes a key epitope implicated in cast nephropathy.**

GP2 and UMOD are structurally related homopolymeric glycoproteins[1] (Extended Data Fig. 1a) that prevent bacterial pathogen adhesion[2,3] and are implicated in multiple pathologies of the intestine and the urinary tract, respectively[4,5]. Recent studies revealed how the C-terminal zona pellucida (ZP) module of UMOD mediates its polymerization[6,7]. However, there is no detailed information on the UMOD N-terminal branch region recognized by FimH[8], suggested to contain a domain with eight cysteines (D8C) conserved in different vertebrate proteins[9], and it is unknown whether the equivalent region of GP2 is also responsible for binding FimH[10].

To address these questions, we first expressed in mammalian cells the whole GP2 branch as well as the corresponding region of UMOD and assessed their ability to selectively capture the lectin domain of FimH (FimH$_L$) from an *Escherichia coli* periplasmic extract. This showed that, as in the case of UMOD, the branch of GP2 is sufficient for interaction with FimH$_L$ (Extended Data Fig. 2).

We then obtained crystals of the GP2 branch, but experimental phasing of its 1.9-Å-resolution data was hindered by relatively high diffraction disorder in one direction and low crystal symmetry. However, molecular replacement with models generated by AlphaFold2 (ref. [11]) allowed us to solve the structure, which was subsequently used to phase two additional crystal forms diffracting to ~1.4 Å resolution (Extended Data Figs. 3 and 4 and Supplementary Table 1). The electron density maps reveal that the GP2 branch is a protein module (henceforth referred to as 'decoy module') that consists of a β-hairpin stabilized by a disulfide bond ($C_x$48-$C_y$59), packed against a globular 'D10C' domain with a new fold including two $3_{10}$ helices, nine β-strands (βA–βI) and five intermolecular disulfides ($C_1$63-$C_8$157, $C_2$85-$C_9$172, $C_3$107-$C_6$145, $C_4$113-$C_{10}$177, $C_5$138-$C_7$146) (Fig. 1a and Extended Data Fig. 1). Notably, the extent of the latter and its $C_1$-$C_8$, $C_2$-$C_9$ disulfides are not compatible

with the original boundaries of the D8C domain[9]; accordingly, GP2 D10C is secreted comparably with the complete branch, whereas a D8C construct is barely expressed and not secreted (Fig. 1b).

The large majority of UMOD pathogenic mutations affect the protein's branch and, in particular, the residues corresponding to the decoy module of GP2 (ref. [4]). Because of 60% sequence identity to UMOD, the crystal structure of the latter immediately explains the effect of many substitutions affecting invariant positions (Fig. 1c–g and Supplementary Table 2). Remarkably, most of these mutations cluster within two structurally important regions of the decoy module, the β-hairpin/D10C domain groove and the disulfide bond-rich region at the opposite end of D10C (Extended Data Fig. 5).

Helical reconstruction of UMOD filaments, together with focused refinement of the protein's branch, recently yielded a composite map of the full-length molecule (Extended Data Fig. 6); however, this information could only be confidently interpreted at the level of the filament core, due to the lack of a reliable model for the branch residues[6]. By combining the crystallographic information on GP2 with AlphaFold2 predictions, we could generate a model of the entire UMOD branch (epidermal growth factor (EGF) domains I–III + decoy module) that was fitted into the cryo-EM density and fused with the coordinates of the filament core to describe the complete protein (Fig. 2a and Supplementary Table 3).

Inspection of the fitted map revealed that, whereas the complex-type carbohydrate linked to D10C N232 (refs. [8,12]) is exposed to the solvent, the high-mannose glycan attached to N275 (refs. [8,12]) emerges from the groove between the β-hairpin and D10C, and packs against the EGF III/β-hairpin junction (Fig. 2b). This suggests that the architecture of the decoy module contributes to maintaining the high-mannose structure of the UMOD N275 glycan, which is crucial for capturing FimH[2,8]. Consistent with this idea, the high-mannose carbohydrate can be fully cleaved by Endoglycosidase H (Endo H) only upon protein denaturation (Fig. 2c). Interestingly, although the GP2 branch also binds FimH$_L$, its D10C domain cannot be glycosylated at the position corresponding to UMOD N275 (R165). However, the presence of a GP2 glycosylation site at N65 (ref. [13])—a residue far away in sequence from R165, but closely located to it within the β-hairpin/D8C groove (Extended Data Fig. 7a)—suggests that this residue may carry a high-mannose glycan equivalent to UMOD N275. In agreement with these considerations, introduction of an N65A mutation in the decoy module of GP2 impairs its interaction with FimH$_L$ (Extended

[1]Department of Biosciences and Nutrition, Karolinska Institutet, Huddinge, Sweden. [2]School of Biological Sciences, Nanyang Technological University, Singapore, Singapore. [3]NTU Institute of Structural Biology, Nanyang Technological University, Singapore, Singapore. [4]US 41-UMS 2014-PLBS, Université de Lille, CNRS, INSERM, CHU Lille, Institut Pasteur de Lille, Lille, France. [5]Department of Biochemistry and Biophysics, Science for Life Laboratory, Stockholm University, Stockholm, Sweden. [6]DeepMind, London, UK. [7]ESRF – The European Synchrotron, Grenoble, France. ✉e-mail: luca.jovine@ki.se

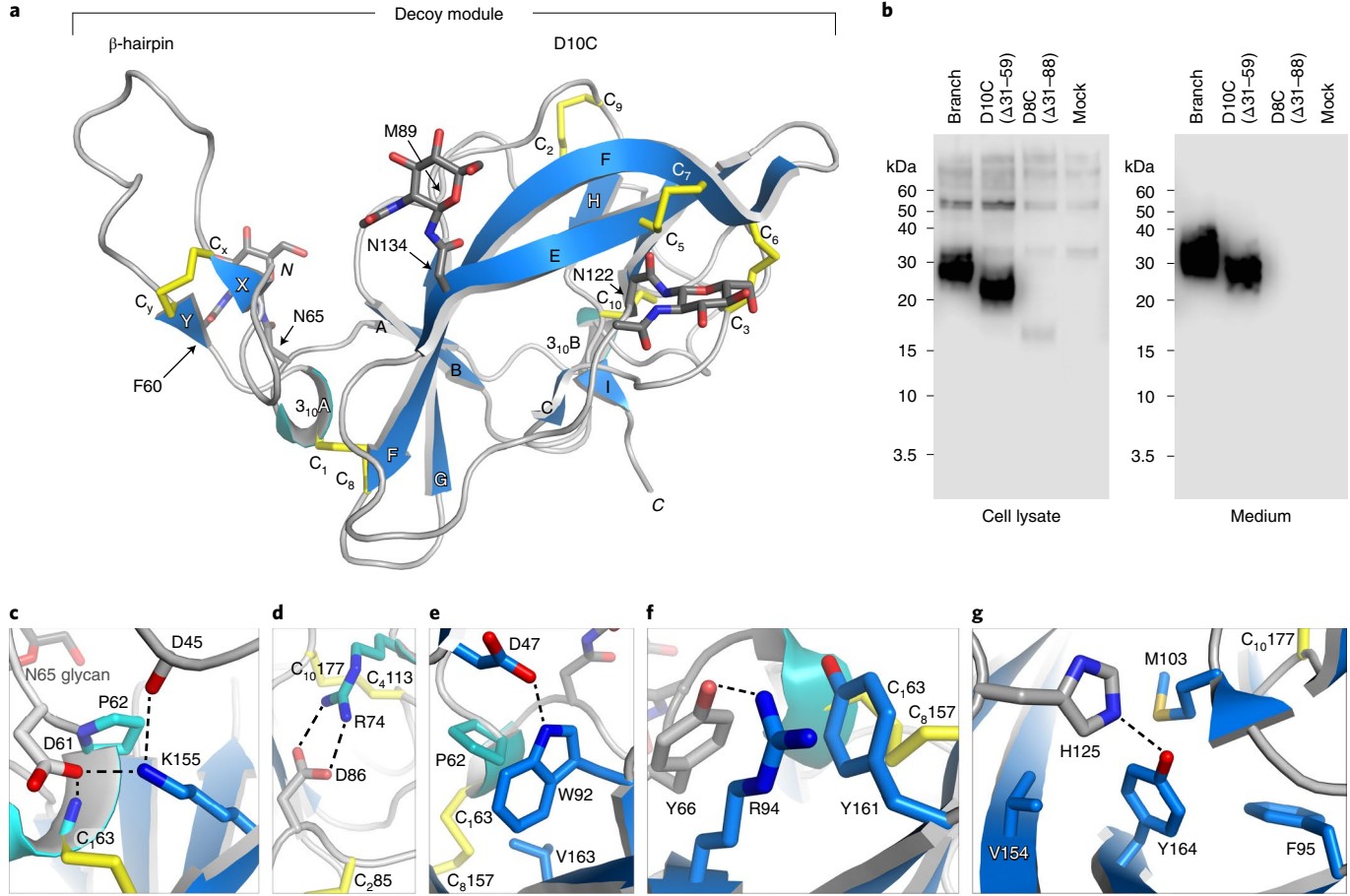

**Fig. 1 | The GP2 branch region includes a D10C domain whose new fold explains patient mutations in UMOD. a**, Overall structure of the GP2 branch region/decoy module, depicted in cartoon representation with β-strands in blue, $3_{10}$ helices in cyan and loops in light gray. Disulfides and glycans are shown as yellow and dark gray sticks, respectively, with oxygen atoms in red and nitrogen atoms in blue. **b**, Reducing western blot comparison of the expression and secretion of GP2 constructs corresponding to the entire branch, D10C or D8C. $n = 3$. **c–g**, Details of the GP2 structure rationalize the effect of kidney disease-associated *UMOD* mutations affecting a set of residues identical between the two proteins (Supplementary Table 2). Selected GP2 D10C domain residues and mutations affecting the corresponding identical residues of UMOD are as follows: GP2 D61, P62, $C_1$63→UMOD D172H, P173L/R, C174R (**c**); GP2 R74, $C_2$85, D86, $C_4$113, $C_{10}$177→UMOD R185C/G/H/L/S, C195F/Y, D196N/Y, C223R/Y, C287F (**d**); GP2 P62, $C_1$63, W92, $C_8$157, V163→UMOD P173L/R, C174R, W202C/S, C267F, V273F/L (**e**); GP2 $C_1$63, R94, $C_8$157→UMOD C174R, R204G/P, C267F (**f**); GP2 Y164, $C_{10}$177→UMOD Y274C/H, C287F (**g**).

Data Fig. 7b) and mass spectrometric analysis of the glycans attached to N65 detects the HexNAc2Hex5 oligomannose structure (Extended Data Fig. 8), indicating that UMOD and GP2 exploit a common molecular strategy to counteract bacterial adhesion.

To gain further insights into this process, which was previously visualized only at low resolution by cryo-electron tomography[8], we reconstituted in vitro the complex between UMOD and $FimH_L$ from uropathogenic *E. coli* (UPEC) UTI89 and studied it by single-particle cryo-EM (Extended Data Fig. 9 and Supplementary Table 3). Despite high conformational variability, this yielded a map with a nominal resolution of 7.4 Å, whose comparison with that of free UMOD showed density for a single copy of $FimH_L$ bound to the D10C region that presents the N275 glycan (Fig. 2d and Supplementary Table 3). Consistent with our binding studies (Extended Data Fig. 2b), the majority of the UMOD/$FimH_L$ interface is clearly made by the decoy module; however, the density of the complex hints at the possibility that the C-terminal region of EGF III may also contribute to the interaction with the lectin.

Finally, our study sheds light on the basis of cast nephropathy, a severe complication of multiple myeloma, by mapping the UMOD epitope recognized by monoclonal light chains/Bence Jones proteins (BJP)[14] to the D10C βE/loop/βF region (Extended Data Fig. 1). Rationalizing previous biochemical studies of this medically crucial interaction[14], the structure suggests that the epitope adopts a rigid conformation stabilized by its involvement in the $C_5$-$C_7$ and $C_3$-$C_6$ disulfides, close proximity to the N232 glycan and hydrophobic interaction with the C terminus of another subunit within the UMOD filament (Fig. 2a,b).

From a general point of view, this work provides an example of how deep learning techniques can substantially aid the X-ray crystallographic and cryo-EM investigation of challenging biological samples, by providing accurate models that can be used to solve the phase problem and aid the fitting of low-resolution density maps, respectively.

## Online content

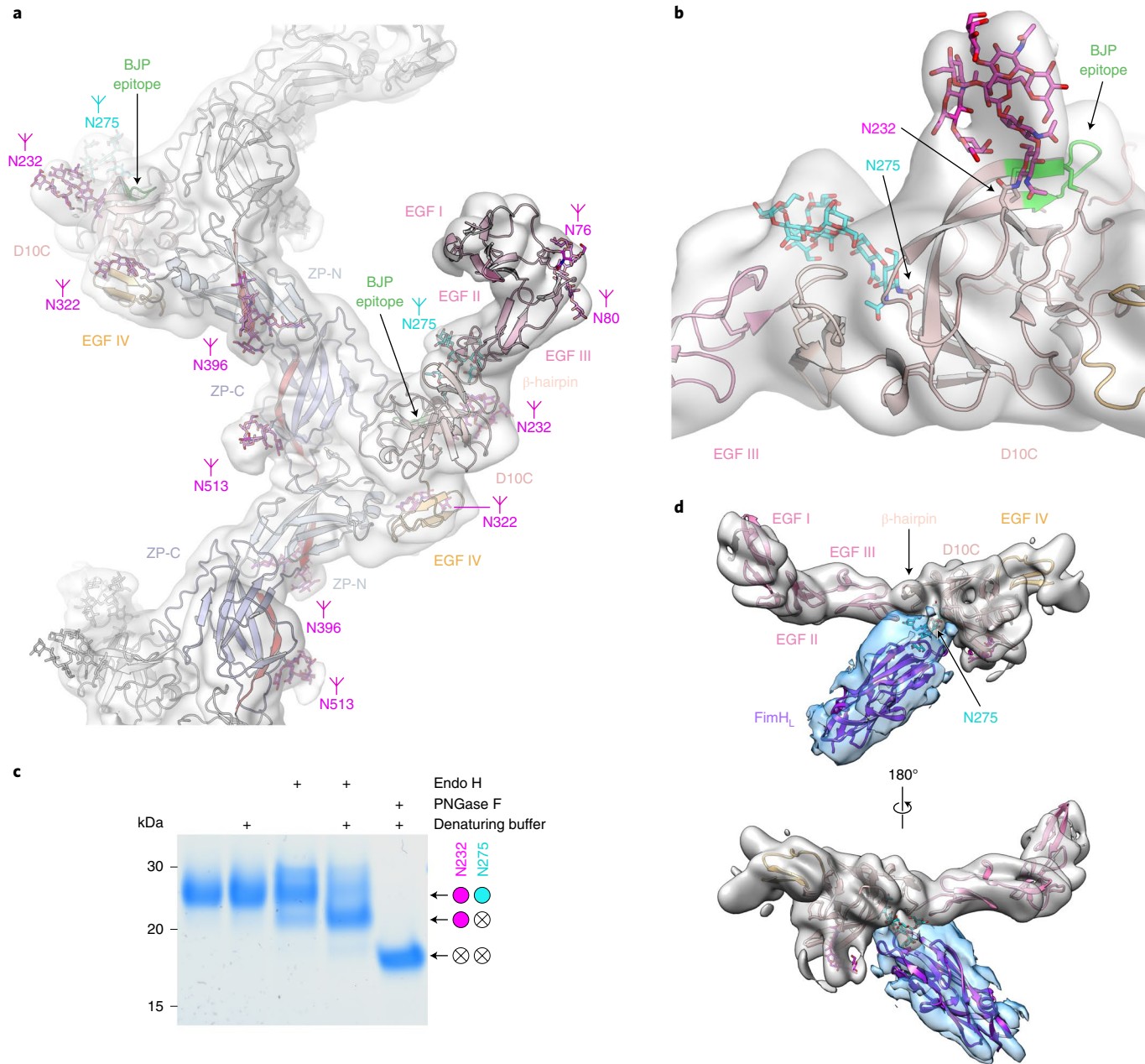

**Fig. 2 | The decoy module fold protects the high-mannose glycan of UMOD and orients it for interaction with bacterial FimH$_L$. a**, Complete atomic model of polymeric UMOD, with *N*-glycans shown as sticks. Elements are colored as in Extended Fig. 1a, with the D10C epitope for BJP in green; additional subunits are gray. **b**, UMOD cryo-EM map region encompassing the protein's decoy module. The Asn side chains carrying the two D10C *N*-glycans and the BJP epitope are indicated. **c**, Consistent with its location within the structure, the N275 high-mannose glycan can be efficiently cleaved by Endo H only in denaturing conditions. Colored circles indicate the presence of the specified glycans, open circles with a cross indicate their absence. *n* = 3. **d**, Recognition of the D10C N275 glycan by the lectin domain of fimbrial adhesin FimH from UPEC UTI89. The cryo-EM map of the UMOD branch + EGF IV is colored gray, the difference map between the densities of the UMOD–FimH$_L$ complex and free UMOD is cyan. PNGase F, Peptide:*N*-glycosidase F.

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

## Methods

**DNA constructs.** Consistent with a cautionary note in UniProt entry P55259 and sequence alignments with homologous sequences from other species, prediction of the signal peptide cleavage propensity of the human GP2 sequence with SignalP[15] suggested that M8, rather than M1, corresponds to the protein's initiator methionine. Moreover, sequence comparisons indicated that GP2 isoform 1 residues V179–S181, which immediately follow the last residue encoded by *GP2* exon 2, are not only absent in isoform α (UniProt P55259-3), but also lack counterparts in human UMOD (UniProt P07911). Based on this information, an open reading frame was designed that encoded GP2α residues M8–S181 (corresponding to isoform 1 residues M8–T178 + D182–S184) followed by a 8× His tag. A corresponding gene and an equivalent UMOD construct, as well as GP2 Δ31-59, Δ31-88 and N65A mutant genes, were also synthesized (GenScript) and all constructs were cloned into pLJ6, a mammalian expression vector derived from pHLsec3 (ref. [16]).

For expressing the *E. coli* FimH lectin domain (FimH$_L$; residues F22–T179), synthetic genes encoding non-tagged and C-terminally His-tagged versions of the protein (including its native signal peptide) were cloned into bacterial expression vectors pD451-SR and pD441-SR/CH (ATUM), respectively.

**Protein expression and purification.** For structural studies, the GP2 branch region was expressed in *N*-acetylglucosaminyltransferase I-deficient Expi293F GnTI- cells (ThermoFisher Scientific), transiently transfected with 25 kDa linear polyethylenimine (Polysciences) as described[17,18]. After capture from the conditioned medium by immobilized metal affinity chromatography (IMAC) and partial deglycosylation with Endo H[19], recombinant GP2 was purified by size-exclusion chromatography (SEC) using a Superdex 75 Increase 10/300 GL column (GE Healthcare) and concentrated to 7 mg ml$^{-1}$ in 20 mM Na-HEPES pH 7.5, 150 mM NaCl.

For evaluation of relative protein secretion levels and FimH$_L$ binding experiments, branch region constructs and mutants thereof were expressed in HEK293T cells[20] grown in DMEM medium supplemented with 4 mM L-Gln, 10% FBS and transiently transfected in 4 mM L-Gln, 2% FBS using 25 kDa branched polyethylenimine (Sigma-Aldrich)[19,21].

For in vitro reconstitution of the UMOD–FimH$_L$ complex, native human UMOD was purified from a healthy 49-year-old male donor using the diatomaceous earth method[22]. His-tagged FimH$_L$ A27V from UPEC strain UTI89 (ref. [23]) was purified by immobilized metal affinity chromatography from the periplasmic extract of *E. coli* OverExpress C43(DE3) cells (Sigma-Aldrich) grown in mannose-free M9 minimal medium. The eluted protein, which was essentially pure by SDS–PAGE analysis, was then dialyzed against 20 mM Na-HEPES pH 7.5, 150 mM NaCl at 0.7 mg ml$^{-1}$ concentration. Finally, purified UMOD and FimH$_L$ were mixed at a molar ratio of 1:3, incubated for 30 min and dialyzed against 10 mM Na-HEPES pH 7.0 (Extended Data Fig. 9).

For binding experiments, a crude periplasmic extract of *E. coli* OverExpress C43(DE3) expressing untagged FimH$_L$ was used (Extended Data Fig. 2a).

**Protein analysis.** Proteins separated by SDS–PAGE were detected with SimplyBlue SafeStain (Invitrogen/ThermoFisher Scientific) or transferred to nitrocellulose membranes (GE Healthcare) for immunoblotting with Penta●His BSA-free anti-5His mouse monoclonal (1:1,000; QIAGEN) and horseradish peroxidase-conjugated goat anti-mouse IgG Fc secondary antibody (1:10,000; Life Technologies/ThermoFisher Scientific). Chemiluminescence detection was performed with Western Lightning ECL Plus (PerkinElmer). Protein deglycosylation under denaturing conditions using either Endo H or Peptide:*N*-glycosidase F (New England Biolabs) was carried out for 1 h at 37 °C, according to the manufacturer's instructions. Gradient gels (4%–12%) were used for SDS–PAGE, except for the deglycosylation experiment shown in Fig. 2c where a 12% gel was used to maximize the separation between bands.

**Protein binding experiments.** Purified C-terminally His-tagged UMOD, GP2 and GP2 N65A decoy module proteins in 20 mM Na-HEPES pH 7.5, 150 mM NaCl (binding buffer) were separately incubated with IMAC beads (GE Healthcare) for 1 h at room temperature. *E. coli* periplasmic extract containing untagged FimH$_L$, adjusted to the binding buffer, was then added and the resulting mixtures were incubated for 2 h at room temperature or overnight at 4 °C. After washing the IMAC beads with binding buffer, bound material was eluted with 20 mM Na-HEPES pH 7.5, 150 mM NaCl, 500 mM imidazole and subjected to SEC as described above. Peak fractions were analyzed by SDS–PAGE, and control SEC runs of the same decoy modules by themselves or a His-tagged version of FimH$_L$ were used to determine the elution volumes of the unbound proteins.

**Protein crystallization.** Crystallization trials of the GP2 branch region, carried out by sitting drop vapor diffusion using a mosquito robot (TTP Labtech), initially yielded triclinic plates that grew in one week at 293K in 25% (v/v) ethylene glycol. After we determined the structure of this crystal form, we obtained two additional forms that also had plate-like morphology but grew at 277K: orthorhombic crystals in 20% (v/v) 1,5-pentanediol, 10% (w/v) PEG 8K, 0.1 M GlyGly/AMPD pH 8.5, 0.5 mM YCl$_3$, 0.5 mM ErCl$_3$, 0.5 mM TbCl$_3$, 0.5 mM YbCl$_3$ (condition E11 of the

MORPHEUS II crystallization screen[24] (Molecular Dimensions)); and monoclinic crystals in 5% (w/v) PEG 20K, 25% (w/v) 1,1,1-tris(hydroxymethyl) propane, 0.1 M MOPSO/bis-tris pH 6.5, 1% (w/v) NDSB-195, 0.01 M spermine, 0.01 M spermidine, 0.01 M 1,4-diaminobutane, 0.01 M DL-ornithine (MORPHEUS II condition H4). Before data collection at synchrotron, crystals were fished directly from the crystallization drops and flash frozen in liquid nitrogen.

**X-ray data collection and reduction.** Datasets for the *P*1, *P*2$_1$2$_1$2$_1$ and *C*2 crystal forms were collected from single specimens at 100 K at European Synchrotron Radiation Facility beamlines ID23-1 (ref. [25]) (λ = 1.0052 Å), ID30B[26] (λ = 0.9763 Å) and ID30A-3 (λ = 0.9677 Å), respectively, using MXCuBE3 (ref. [27]). All data was processed with XDS[28] (Supplementary Table 1), with high-resolution data cutoffs chosen on the basis of statistical indicators CC$_{1/2}$ and CC*[29,30]. Although the *P*1 crystals diffracted reproducibly to better than 3.0 Å resolution, a single specimen yielded data extending well beyond a Bragg spacing of 2.0 Å; unfortunately, probably because of the disorder, the diffraction extent of this particular crystal was severely underestimated by the data collection strategy software, so that we were only able to process the resulting data to 1.9 Å.

**Experimental phasing attempts.** Despite the workable resolution of its diffraction, the *P*1 crystal form suffered from disorder parallel to the *b*c* planes, that is reflected by relatively high $R_{merge}$ and $R_{meas}$ values. Although this did not prevent us from ultimately solving the structure by molecular replacement (MR), it precluded multiple attempts to phase the data experimentally by sulfur-single wavelength anomalous dispersion. Parallel attempts to obtain usable derivative data from crystals soaked with Pt or Au compounds also failed, because of the apparent lack of specific binding sites for these heavy atoms. Similarly, no heavy atom bound to the *C*2 crystal form of the protein despite the fact that this was obtained in the presence of a mixture of different lanthanides and yttrium.

**Structure solution by molecular replacement with AlphaFold2 models.** AlphaFold2 (AlphaFold Monomer 2.0)[11] was used to generate five independent models of residues V29–S181 of GP2α, with relative r.m.s. deviations (r.m.s.d.) of 0.6–1.7 Å. After removal of a low-confidence N-terminal region (residues V29–L44), visual inspection of the models suggested further trimming to residues D61–S181, which clearly belonged to a single globular domain (Extended Data Fig. 3a). The resulting coordinate sets (r.m.s.d. 0.1–0.2 Å), with per-residue pseudo-*B* factors corresponding to 100-(per-residue confidence (pLDDT[11])), were combined into an ensemble that was used to phase the *P*1 data by MR with Phaser[31]. Using a search model r.m.s.d. variance of 1 Å, this found a single solution consisting of two molecules per asymmetric unit (LLG 1258, TFZ 31.6), whose correctness was readily confirmed by initial refinement (*R* 0.31, $R_{free}$ 0.36) and positive difference density for the *N*-acetylglucosamine (GlcNAc) residues attached to GP2 N65, N122 and N134 as well as part of the β-hairpin (Extended Data Fig. 3b,c). After one round of autobuilding in PHENIX[32], the structure was completed by alternating manual rebuilding in Coot[33] and ISOLDE[34] with refinement using phenix.refine[35]. Protein geometry and carbohydrate structure validation was carried out with MolProbity[36] and Privateer[37], respectively, and data reduction, refinement and validation statistics calculated using phenix.table_one[38] are reported in Supplementary Table 1. Because of a lack of density for the residues making up the loop of the β-hairpin, the final model consists of GP2 residues S41–G49 and H57–S181, as well as five GlcNAc residues attached to N65, N122 (chains A and B) and N134 (chain A only). Using these coordinates as a reference, the top ranked AlphaFold2 model had a Global Distance Test (GDT_TS) score of 94.9 (or 97.2 if only the D10C domain is considered).

An ensemble of the two chains of a partially refined model of the *P*1 structure was used to phase the *P*2$_1$2$_1$2$_1$ data (with one molecule in the asymmetric unit) by MR (LLG 8167, TFZ 41.7; initial *R* 0.23, $R_{free}$ 0.25); residues D61–S181 of the refined *P*2$_1$2$_1$2$_1$ model were in turn used for MR phasing of the *C*2 data (LLG 8539, TFZ 82.9; initial *R* 0.24, $R_{free}$ 0.25). As expected on the basis of the *P*1 MR results, both the orthorhombic and monoclinic structures could, in principle, also have been solved using the initial AlphaFold2 ensemble (*P*2$_1$2$_1$2$_1$: LLG 1325, TFZ 33.5; initial *R* 0.32, $R_{free}$ 0.35; *C*2: LLG 1232, TFZ 31.9; initial *R* 0.32, $R_{free}$ 0.34). After rebuilding, refinement and validation as described for the *P*1 crystal form, the final *P*2$_1$2$_1$2$_1$ and *C*2 models contain amino acids Y42–S181 and L44–S181, respectively, as well as two GlcNac residues attached to N65 and N122; in addition, the orthorhombic model includes two residues belonging to the C-terminal His-tag, whereas the monoclinic one contains the GlcNac attached to N134.

**Cryo-EM data collection.** Data collection and processing details for full-length native human UMOD have been reported[6].

For collecting cryo-EM data from the UMOD–FimH$_L$ complex (Supplementary Table 3), prepared as described in the section 'Protein expression and purification,' the specimen (1.8 mg ml$^{-1}$) was applied in 3-μl volumes onto glow-discharged Cu R2/2 holey carbon 300 mesh grids (Quantifoil). After blotting for 2 s, grids were plunged into liquid ethane cooled by liquid nitrogen using a Vitrobot Mark IV (ThermoFisher Scientific). Cryo-EM experiments were performed at the Cryo-EM Swedish National Facility, SciLifeLab, Stockholm. Videos were collected using fringe-free imaging and aberration-free image shift with the EPU data acquisition

software, on a Titan Krios electron microscope (ThermoFisher Scientific) operated at 300 kV, using a K3 camera equipped with a BioQuantum energy filter (Gatan-Ametek). Videos were taken at ×105,000 nominal magnification in counting mode with a dose rate of 15 e px$^{-1}$ s$^{-1}$ and a total dose of 40 e/Å$^2$ distributed over 40 subframes, gain-corrected and then compressed using video compression in RELION[39]. Motion correction with dose weighting was also performed in RELION[40] within the Scipion software suite[41].

**Cryo-EM data processing.** Processing of the cryo-EM data of the UMOD–FimH$_L$ complex followed the general workflow used for reconstructing the full-length UMOD filament[6]. First, contrast transfer function determination was carried out using CTFFIND in RELION. An in-house script designed specifically for filament picking (Cryo-EM-filament-picker)[42] was then used to select end-to-end filament coordinates. After two-dimensional classification in cryoSPARC[43], selected particle coordinates were transferred back to RELION for three-dimensional (3D) classification, 3D helical refinement, particle subtraction and final non-helical refinement and polishing. Specifically, starting from a total of 13,616 raw micrographs, 3,767,790 particles (helical segments with 70 Å step size) were auto-picked and extracted on the basis of motion correction and contrast transfer function estimation; based on two-dimensional classification quality evaluated with cryoSPARC, a subset of 1,139,808 particles was then selected for further processing. Because FimH$_L$ occupancy varied among filaments, segments with higher FimH$_L$ occupancy were selected during iterative RELION 3D classification runs. Finally, 225,819 homogeneous particles were subjected to auto-refinement and postprocessing. To improve the local density of the FimH$_L$-binding region, we performed particle subtraction to mask out the UMOD helical core and continued local refinement in RELION. Ultimately, a density representing the UMOD branch–FimH$_L$ complex with an overall average resolution of 7.4 Å was obtained by auto-refining the subtracted particles with a UCSF Chimera[44]-generated mask that only covered the binding region (Extended Data Fig. 9 and Supplementary Table 3).

**Cryo-EM map fitting, model refinement and validation.** A complete atomic model of full-length UMOD was assembled in several steps. First, five independent models of the whole UMOD branch (residues D25–S191) were generated with AlphaFold2; all these models shared the same domain boundaries, fold and disulfide connectivity, with their overall r.m.s.d. (0.4–4.3 Å) simply reflecting differences in the orientation of EGF I–III (r.m.s.d. 0.2–0.4 Å) relative to the decoy module (r.m.s.d. 0.1–0.2 Å). Second, although the overall r.m.s.d. values between the AlphaFold2 models of the GP2 D10C domain and the corresponding experimental structures (average ~0.5 Å) were not much larger than those between the latter (average 0.1 Å), local differences could be observed at the level of the relatively flexible 3$_{10}$B/βB loop as well as a subset of side chains. To consider these alternatives while fitting the cryo-EM density of the UMOD D10C domain (62% sequence identical to that of GP2), the $P2_12_12_1$ and $C2$ high-resolution structures of GP2 D10C were each used to generate five homology models of UMOD D10C using MODELLER[45]. The respective models with the best Discrete Optimized Protein Energy (DOPE) scores[46] were then used as starting points for exploring different possible conformations by molecular dynamics in YASARA Structure[47]. Third, the top AlphaFold2 model and $P2_12_12_1$/$C2$-structure derived homology models (r.m.s.d. 0.7/0.8 Å) of D10C were individually rigidly docked with UCSF Chimera into the 3D reconstruction of full-length UMOD (overall nominal resolution 4.7 Å)[6], whose masking and postprocessing with RELION was optimized to obtain the best possible density for the D10C-containing region near the center of the map. The resulting map fit correlations of the AlphaFold2 model and the homology models were 0.884 and 0.892/0.896, respectively. Fourth, the placed AlphaFold2 model was locally rebuilt, taking into account—if available— alternative possibilities suggested by the superimposed homology models. At this stage, we also connected the C terminus of D10C to the N terminus of the atomic model of the UMOD filament core (PDB ID 6TQK)[6], consisting of the EGF IV domain and the ZP module (Extended Data Fig. 1a); rebuilt the C-terminal end of the ZP-C domain interacting with D10C[6]; and built the glycan chains attached to N232 and N275. The resulting coordinates were then subjected to global real-space and group ADP refinement in PHENIX[48], essentially as described[6] (CC$_{mask}$ 0.74; CC$_{box}$ 0.79; CC$_{peaks}$ 0.39; CC$_{vol}$ 0.72; mean CC$_{carbohydrates}$ 0.62). Finally, the model was completed by fusing it with EGF I–III/β-hairpin coordinates extracted from the top AlphaFold2 model of the whole UMOD branch, flexibly fit into a cryo-EM map of the same protein region (overall nominal resolution 6.1 Å) using Namdinator[49] (CC$_{mask}$ 0.59; CC$_{box}$ 0.76; CC$_{peaks}$ 0.43; CC$_{vol}$ 0.56; mean CC$_{carbohydrates}$ 0.60). Following further rebuilding and real-space refinement against a composite map of full-length UMOD generated by multibody refinement[6] (Extended Data Fig. 6), performed using the starting model as a reference for generating torsion restraints, protein geometry and carbohydrate structure were validated using PHENIX[50]/MolProbity (Supplementary Table 3) and Privateer; model-to-map validation was carried out with PHENIX (CC$_{mask}$ 0.75; CC$_{box}$ 0.81; CC$_{peaks}$ 0.48; CC$_{vol}$ 0.73; mean CC$_{carbohydrates}$ 0.77). The final model consists of 1,127 protein residues, corresponding to a complete chain (chain A, D25–F587) and two half chains (chain B, S444–F587; chain C, D25–S444) that together recapitulate all the protein-protein interactions in the UMOD filament, as well as 84 N-glycan residues.

The model of the UMOD branch + EGF IV/FimH$_L$ complex was generated by manually docking the crystallographic structure of FimH$_L$ bound to trimannose (chains A and F of PDB ID 6GTW)[51] into the difference density between the cryo-EM maps of the FimH-bound and free UMOD branch + EGF IV (calculated using TEMPy:DiffMap[52] and masked around the decoy module region), so that the lectin made an equivalent interaction with the α1,3 branch of the high-mannose glycan attached to UMOD N275. After optimizing the position of FimH$_L$ against the difference map by rigid-body refinement, introducing A27V, S62A substitutions to match the sequence of FimH from UPEC UTI89 variant A27V and rebuilding the other residues of the N275 glycan, the whole complex was finally subjected to real-space refinement with restraints generated using the starting coordinates as a reference (Supplementary Table 3).

**Sequence-structure analysis.** Structure-based sequence alignments, generated using MAFFT[53] as implemented in ConSurf[54], were rendered with ESPript[55]. For calculating consensus information at different thresholds, a ConSurf alignment that sampled homologs of the GP2 branch domain with 35–95% identities was first pruned of incomplete sequences (yielding a final set of 129 aligned sequences) and then processed with MView[56].

GDT_TS scores were calculated using the AS2TS server[57] and possible structural similarities were assessed using Dali[58]. Secondary structure was assigned using STRIDE[59]; structural figures were generated with PyMOL (Schrödinger, LLC) and UCSF Chimera/ChimeraX[60].

**Site specific N-glycosylation analysis by liquid chromatography–tandem mass spectrometry.** The His-tagged GP2 branch region purified from the conditioned medium of HEK293T cells was denatured, reduced and alkylated before digestion with either sequencing-grade AspN or with pepsin/chymotrypsin. The digests were analyzed on an Ultimate 3000 nanoLC system online coupled to a QExactive mass spectrometer (ThermoFisher Scientific). Raw data was analyzed by ByonicTM (Protein Metrics Inc.) set to identify glycopeptides from the fragmented parent ion. The acceptance criterion was a false discovery rate on the protein level below 1%. Peptide and glycan sequences were analyzed by ByonicTM from the higher-energy C-trap dissociation (HCD) spectra and verified manually.

**Reporting Summary.** Further information on research design is available in the Nature Research Reporting Summary linked to this article.

## Data availability

The UniProt (https://www.uniprot.org/) IDs for hGP2 and hUMOD are P55259 and P07911, respectively; the IDs of other sequences reported in the alignment of Extended Data Fig. 1b are Q9D733 (mGP2), Q91X17 (mUMOD), Q8WWZ8 (hLZP), Q8R4V5 (mLZP), Q8N2E2 (hVWDE) and Q6DFV8 (mVWDE). The Electron Microscopy Data Bank (EMDB; https://www.ebi.ac.uk/emdb/) ID of the UMOD filament map used for assembling the composite map shown in this work is EMD-10553; the UMOD filament core and FimH$_L$/trimannose coordinates used as starting models can be retrieved from the Protein Data Bank (PDB; https://www.rcsb.org/) with IDs 6TQK and 6GTW, respectively. Structure factors and atomic models for the $P1$, $P2_12_12_1$ and $C2$ crystal forms of the GP2 decoy domain have been deposited in the PDB with accession codes 7P6R, 7P6S and 7P6T, respectively. Cryo-EM density maps of full-length UMOD and the UMOD branch + EGF IV/FimH$_L$ complex have been deposited in the EMDB with accession codes EMD-13378 and EMD-13794, respectively; the corresponding coordinates have been deposited in the PDB with accession codes 7PFP and 7Q3N. Source data are provided with this paper.

## Code availability

The Python code for filament picking is available at: https://doi.org/10.5281/zenodo.5807535.

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

## Acknowledgements

We thank D. Briggs (The Francis Crick Institute, London) for advice on transient transfection of Expi293F cells; A. Vegvari (Karolinska Institutet Proteomics Biomedicum core facility) for the MS analysis of the FimH$_L$ bands; the Plateforme d'Analyses des Glycoconjugués (PAGés) and the Plateforme d'Analyse Protéomique et de Protéines Modifiés (P3M) for GP2 N65 glycan LC-MS/MS; the staff of the European Synchrotron Radiation Facility (ESRF; Grenoble) and the Swedish National Cryo-EM Facility (Stockholm) for help with X-ray and cryo-electron microscopy data collection and preprocessing; A. Zemla (Lawrence Livermore National Laboratory, Livermore) for help with GDT_TS calculations; and T. Terwilliger (New Mexico Consortium, Los Alamos) for discussion. This work was supported by the Swedish Research Council (project grants 2016-03999 and 2020-04936 to L.J.), the Karolinska Institutet Research Foundation (grant 2016fobi50035 to L.J.), the Knut and Alice Wallenberg Foundation (project grant 2018.0042 to L.J.) and the Ministry of Health, Singapore, NMRC grant (MOH-000382-00 to B.W.).

## Author contributions

A.S., S.N. and L.H. expressed and purified proteins. A.S. and S.N. carried out protein-protein interaction experiments. A.S., L.J. and D.d.S. performed crystallographic research. K.T. and J.J. generated AlphaFold2 models. C.X., B.W., L.J., M.C. and A.S. performed cryo-EM research. N.Y. analyzed protein glycosylation by mass spectrometry. L.J. coordinated the study and wrote the manuscript with A.S., based on input from all other coauthors.

## Funding

## Competing interests

J.J. has filed provisional patent applications relating to machine learning for predicting protein structures. The other authors declare no competing interests.

## Additional information

**Extended data** is available for this paper at https://doi.org/10.1038/s41594-022-00729-3.

**Correspondence and requests for materials** should be addressed to Luca Jovine.

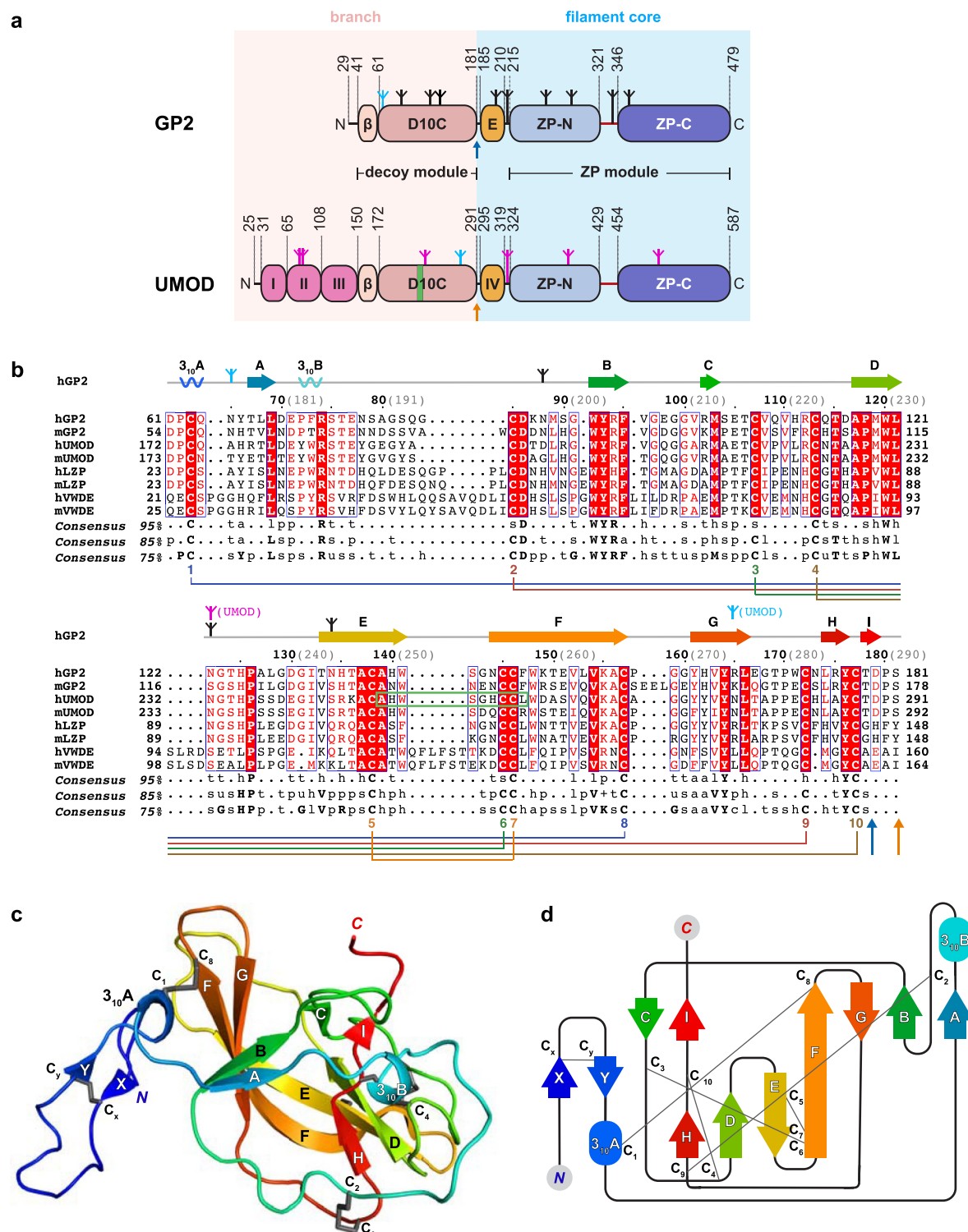

**Extended Data Fig. 1 | See next page for caption.**

**Extended Data Fig. 1 | Structure of the GP2 N-terminal branch and its relation with the corresponding regions of UMOD and additional mammalian proteins. a**, Domain architecture of mature human GP2 and UMOD. Domains are indicated by their acronyms, except for UMOD epidermal growth factor (EGF) domains that are labeled according to their roman number, the single EGF domain of GP2 (corresponding to UMOD EGF IV) that is labeled as 'E' and the β-hairpin of the decoy module ('β'). The UMOD D10C epitope recognized by Bence-Jones proteins (BJP)[14] is shown as a green stripe. Black and magenta inverted tripods indicate the *N*-glycosylation sites of GP2 and UMOD, respectively, with the high-mannose chains attached to GP2 N65 (this study) and UMOD N275[8,12] colored cyan. The position corresponding to the alternative 3' splice site generating the β isoform of GP2 (T178|D179)[61] and the elastase cleavage site of UMOD (S291|S292)[62] are indicated by vertical blue and orange arrows, respectively. **b**, Alignment of D10C domain sequences from human (h) and murine (m) homologues of GP2 and UMOD, as well as liver-specific zona pellucida protein (LZP/OIT3, a molecule that can also interact with UMOD in the kidney and urine[63]) and von Willebrand factor D and EGF domain-containing protein (VWDE; a protein involved in appendage regeneration in a variety of vertebrate species[64]). Identical residues are highlighted in white and shaded in red; conserved residues are red and marked by blue frames when clustered. Consensuses at different sequence identity thresholds, based on a comprehensive alignment of homologous sequences, are also reported (bold uppercase characters: amino acids with the same one-letter code; regular lowercase characters: l, [I,V,L]; h, [F,Y,W,H,I,V,L]; +, [H,K,R]; -, [D,E]; p, [Q,N,S,T,C,H,K,R,D,E]; u, [G,A,S]; s, [G,A,S,V,T,D,N,P,C]; t, [G,A,S,Q,N,S,T,C,H,K,R, D,E]; (.), any amino acid). GP2 secondary structure elements, rainbow-colored from blue (N-terminus) to red (C-terminus), and disulfide bond connectivity are shown above and below the alignment, respectively. Other elements are labeled as in (a), with a green box indicating the BJP epitope[14]. Black bold numbers above the alignment indicate hGP2 residues; light grey numbers between parentheses refer to the corresponding hUMOD residues. **c**, Cartoon representation of the GP2 decoy module, rainbow-colored following the same scheme used for the secondary structure elements of (b). Disulfide bonds are represented as grey sticks. **d**, Topology and disulfide connectivity diagram of the decoy module.

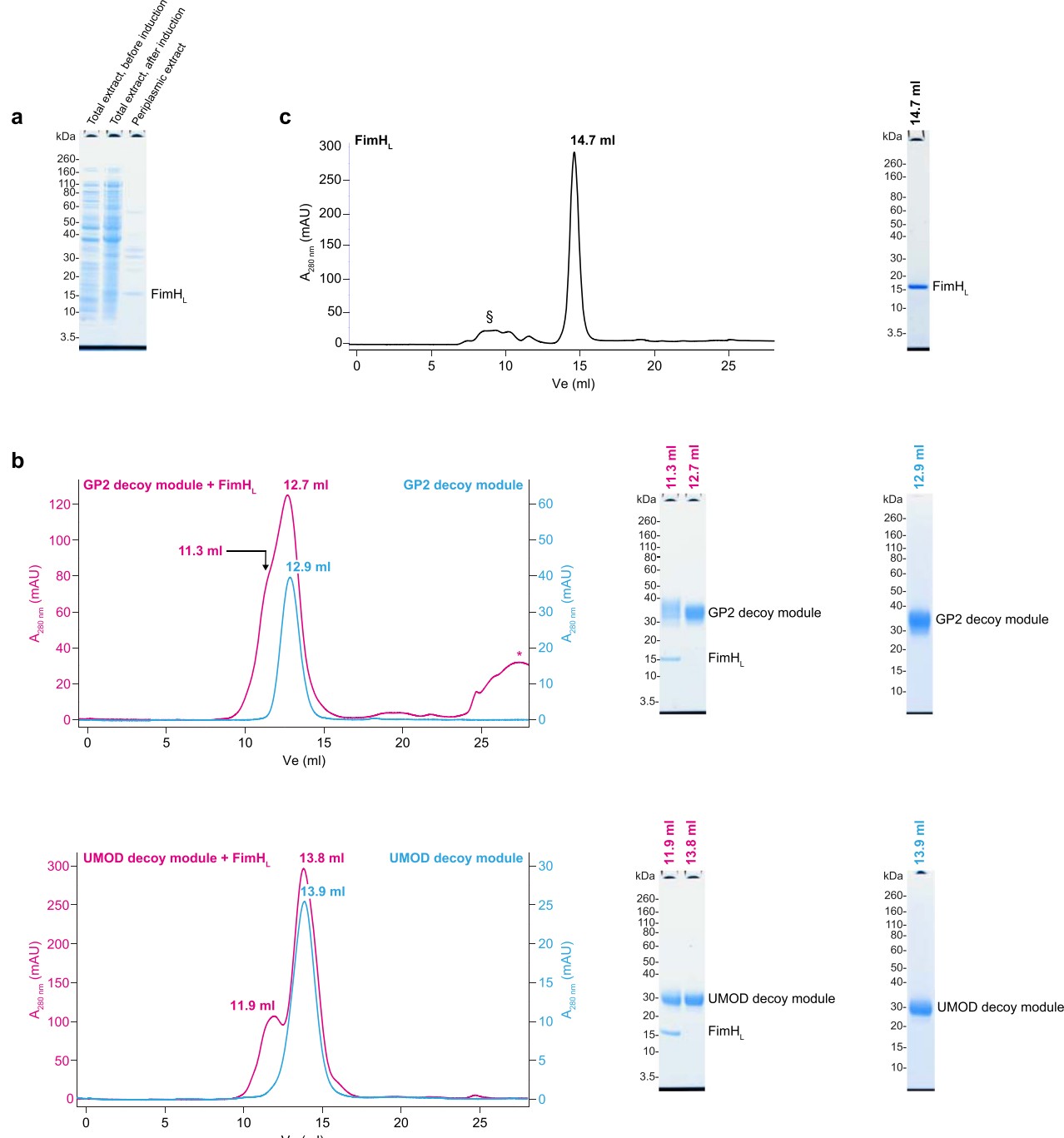

**Extended Data Fig. 2 | The isolated GP2 branch and the corresponding decoy module of UMOD bind FimH$_L$. a**, For assessing whether the lectin domain of FimH is able to bind *in vitro* to the branch of GP2 or the equivalent region of UMOD (corresponding to the respective decoy modules, see main text), untagged FimH$_L$ was expressed in *E. coli* and a crude periplasmic extract was prepared. $n = 2$. **b**, SEC analysis of the material eluted after incubating purified His-tagged GP2 or UMOD decoy modules bound to IMAC beads with the FimH$_L$-containing *E. coli* periplasmic extract (magenta curves). In both cases, reducing SDS-PAGE of peak fractions and tandem mass spectrometry (MS/MS) of the corresponding ~15 kDa bands show the presence of complexes between the decoy modules and the bacterial adhesin, indicating that the former are able to selectively recognize the latter among the pool of periplasmic proteins. SEC elution profiles of the GP2 and UMOD decoy domains by themselves are also shown (light blue curves), and a low-molecular weight contaminant peak is indicated by *. GP2 decoy module, UMOD decoy module: $n = 3$; GP2 decoy module/FimH$_L$, UMOD decoy module/FimH$_L$, $n = 2$. **c**, Control SEC profile of unbound His-tagged FimH$_L$ with SDS-PAGE analysis of the peak. § indicates minor high-molecular weight contaminants eluting with or close to the void volume. $n = 3$.

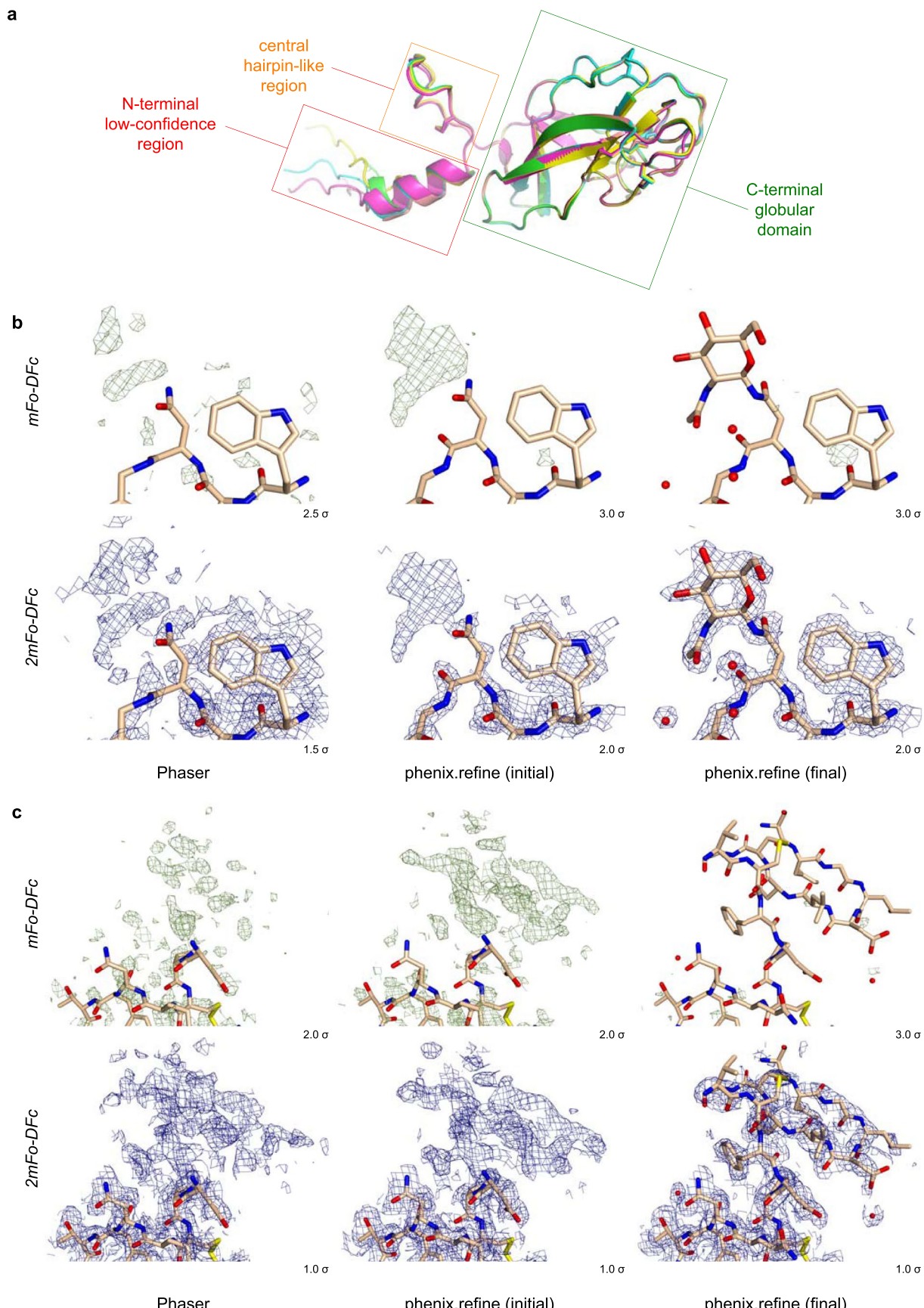

**a**

central hairpin-like region

N-terminal low-confidence region

C-terminal globular domain

**b**

*mFo-DFc*

2.5 σ          3.0 σ          3.0 σ

*2mFo-DFc*

1.5 σ          2.0 σ          2.0 σ

Phaser          phenix.refine (initial)          phenix.refine (final)

**c**

*mFo-DFc*

2.0 σ          2.0 σ          3.0 σ

*2mFo-DFc*

1.0 σ          1.0 σ          1.0 σ

Phaser          phenix.refine (initial)          phenix.refine (final)

**Extended Data Fig. 3 | See next page for caption.**

**Extended Data Fig. 3 | AlphaFold2 model phasing of the GP2 branch *P*1 X-ray data. a**, Superposition of five AlphaFold2 models of the GP2 N-terminal branch indicates the presence of three distinct units, with a central hairpin-like region (residues D45-F60; orange box) separating an N-terminal low-confidence region (residues V29-L44; red box) from a C-terminal globular domain (residues D61-S181; green box). An ensemble corresponding to the latter was used as search model for MR. **b-c**, Electron density for an Endo H cleavage-derived N-acetylglucosamine residue attached to N122 (b) and the hairpin region (c), two GP2 elements not included in the MR search ensemble. Fourier maps at different stages of the structure determination process are shown, contoured at the indicated levels.

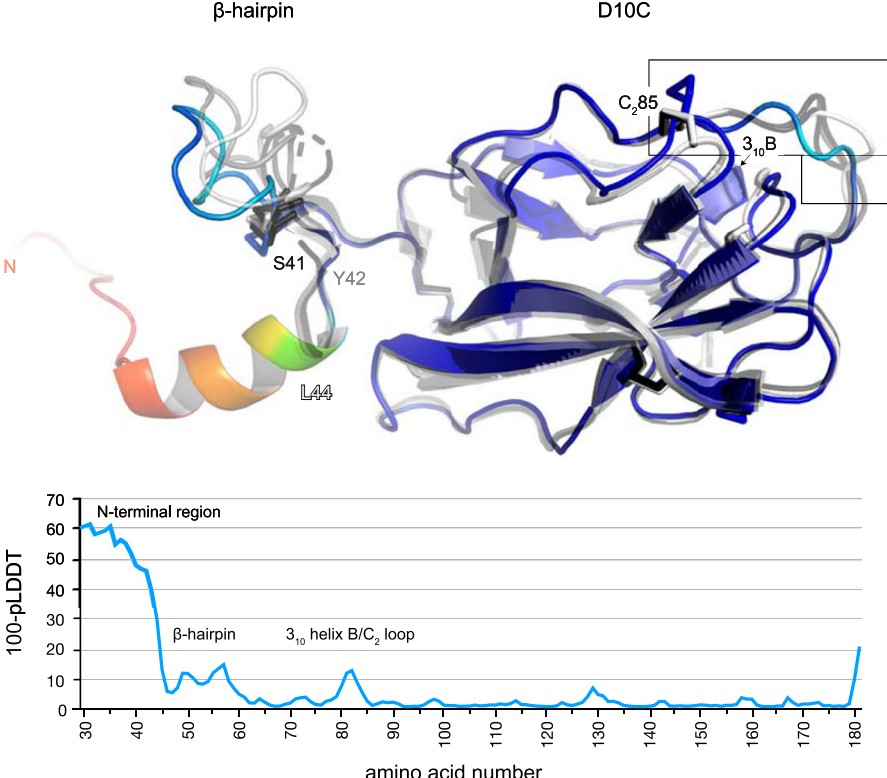

**Extended Data Fig. 4 | Comparison of the predicted and experimental structures of the human GP2 branch region.** The crystallographic models, shown as semi-transparent cartoons colored in black (*P1*), grey (*P2₁2₁2₁*) and white (*C2*), are superimposed on the top AlphaFold2 model, colored from blue to red according to a 100-(per-residue confidence (pLDDT[11])) scale that ranges from 0 (blue; maximum confidence) to 100 (red; minimum confidence). Note how the low-confidence prediction for the N-terminal region of the GP2 branch matches the observations that the corresponding residues are largely structurally disordered in the different crystal forms of the protein (whose first resolved residues, S41/Y42 (*P1* chains A/B), Y42 (*P2₁2₁2₁*) or L44 (*C2*) are indicated) and apparently proteolytically removed from mature native GP2[65]. Similarly, two protein regions that display relative structural flexibility in the GP2 crystals, the β-hairpin and part of the long loop connecting 3₁₀ helix B to conserved Cys 2 (white box), contain residues predicted with lower confidence by AlphaFold2.

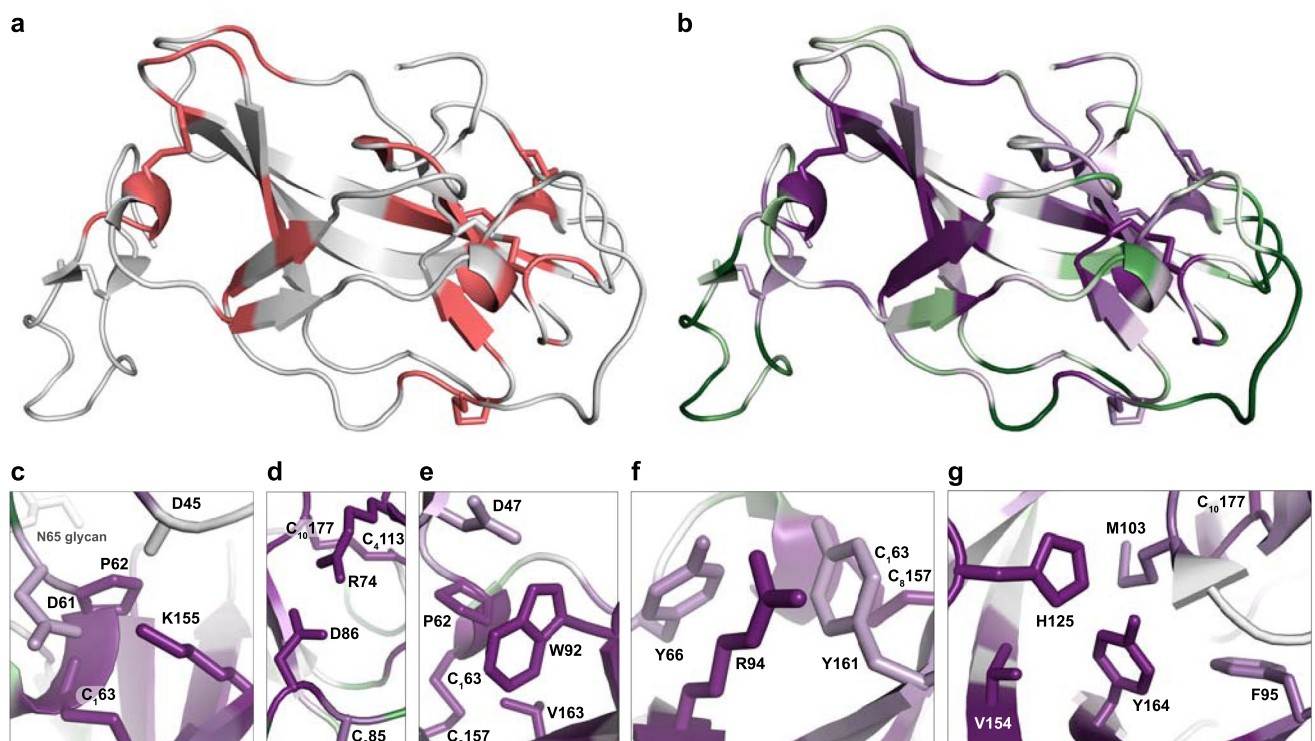

**Extended Data Fig. 5 | Pathogenic substitutions in the D10C domain affect clusters of highly conserved residues. a-b**, GP2 D10C residues corresponding to UMOD amino acids mutated in kidney disease patients (panel a, red) are largely clustered into two highly conserved protein regions (panel b). Sequence conservation is represented using a color spectrum ranging from green (lowest conservation) to violet (highest conservation). **c-g**, Alternative representation of the structural details shown in Fig. 1c–g, with residues colored by sequence conservation.

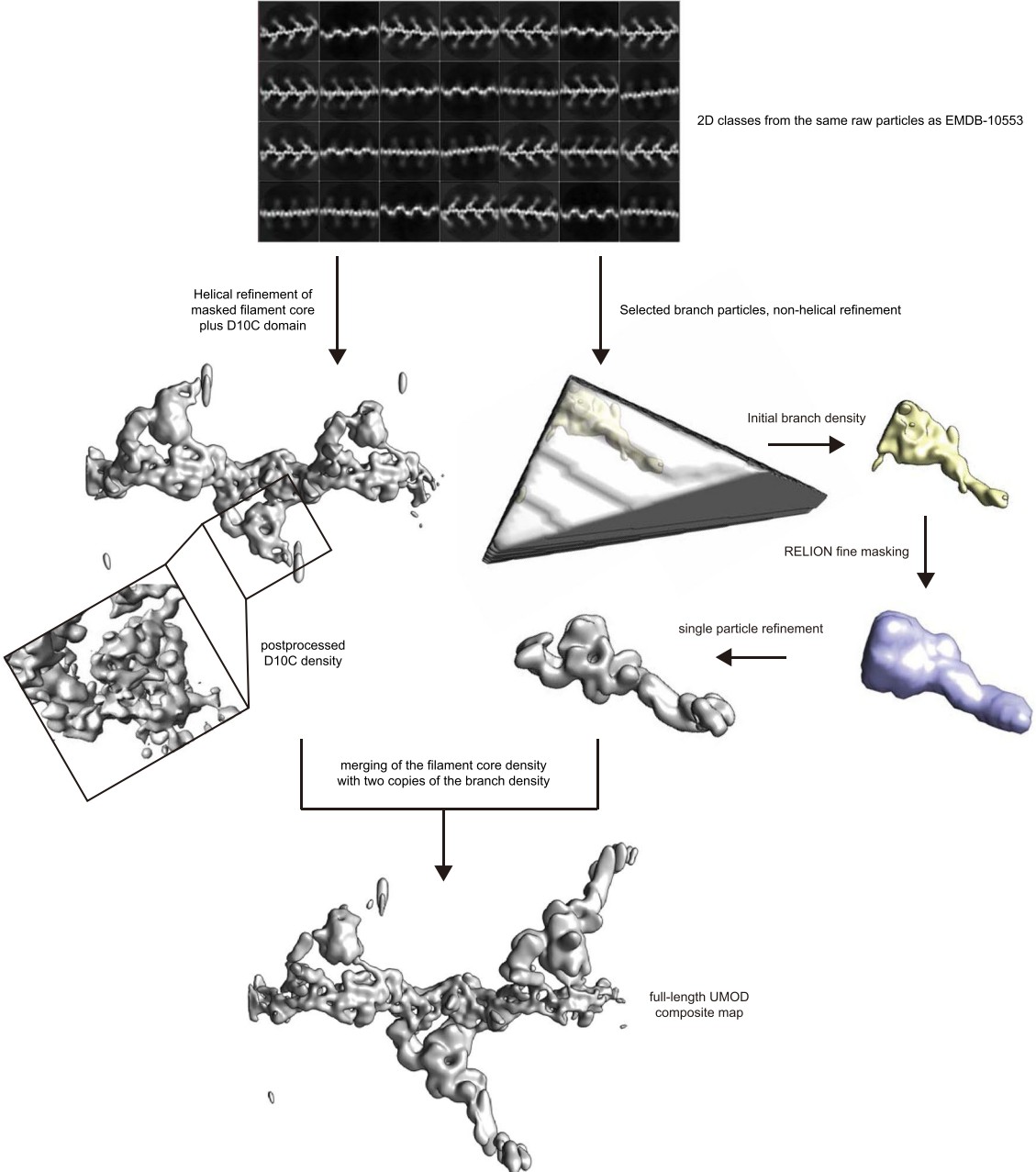

**Extended Data Fig. 6 | Assembly of the composite map of full-length UMOD.** Multi-body refinement of the UMOD filament core plus D10C domain (left path) and the whole UMOD branch (right path) were performed separately. Helical symmetry was applied to the filament core plus D10C, after the best homogenous filamentous segments were selected based on 2D classes. Meanwhile, the particles with the better contrast, more extended branch features were independently selected, locally 3D classified and refined, without helical symmetry. The final composite map was assembled by merging copies of the branch with the filament core plus D10C.

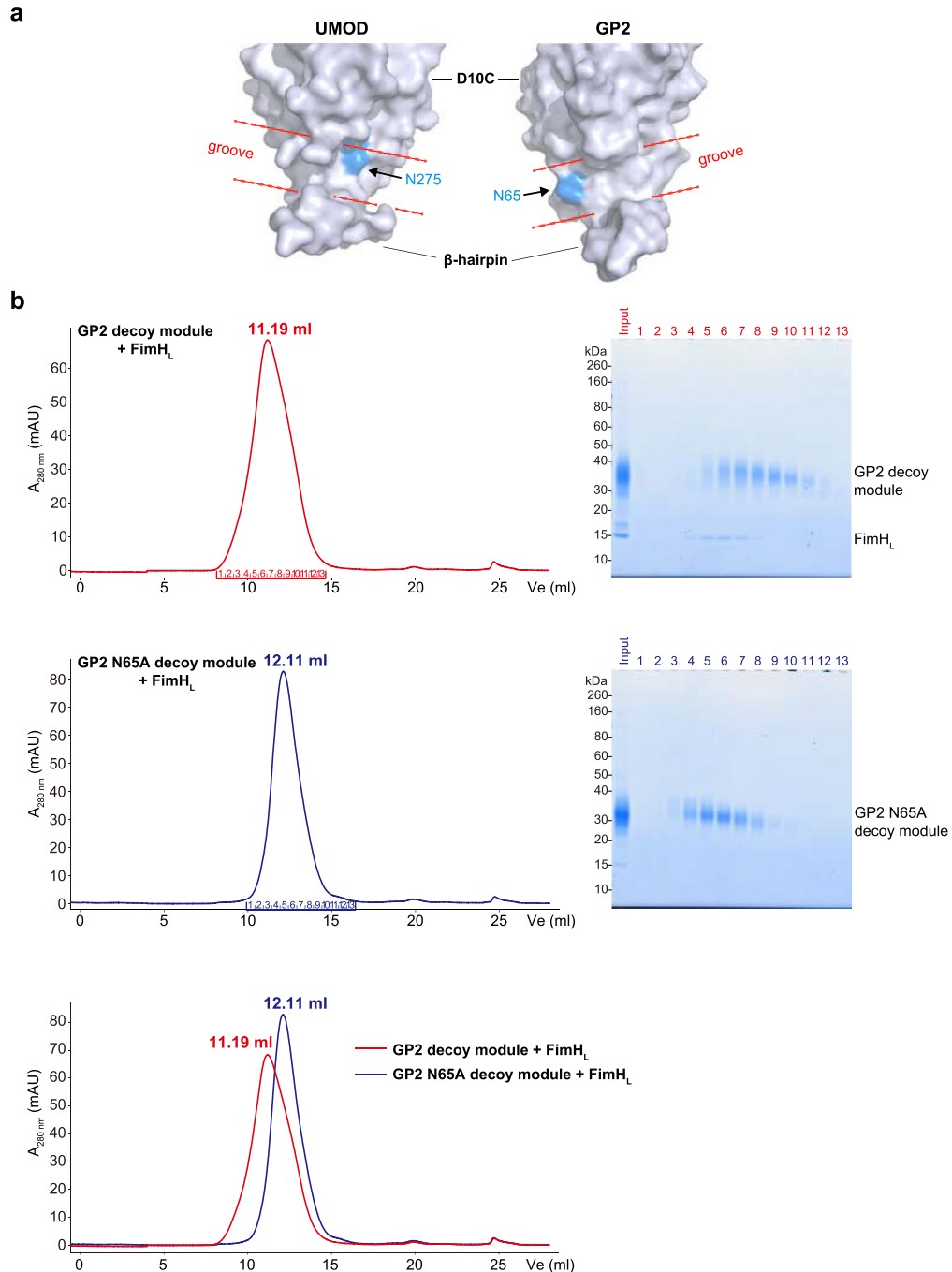

**Extended Data Fig. 7 | Inactivation of the N65 glycosylation site of GP2 impairs the interaction between the protein's decoy module and FimH$_L$.**
**a**, The FimH-binding high-mannose glycan attached to UMOD N275 is located in the groove between the β-hairpin and D10C domain moieties of the protein's decoy module (left panel). Although this sequon is not conserved in the decoy module of GP2, the groove of the latter contains a different, but closely spaced, N-glycosylation site at position 65 (right panel). **b**, SEC analysis of the material eluted after incubating an *E. coli* periplasmic extract containing untagged FimH$_L$ with wild-type or N65A mutant GP2 decoy modules immobilized on IMAC beads (left panels). Reducing SDS-PAGE analysis of the corresponding peak fractions (right panels) shows that FimH$_L$ binds to the wild-type GP2 decoy module but not to the N65A mutant. $n = 2$.

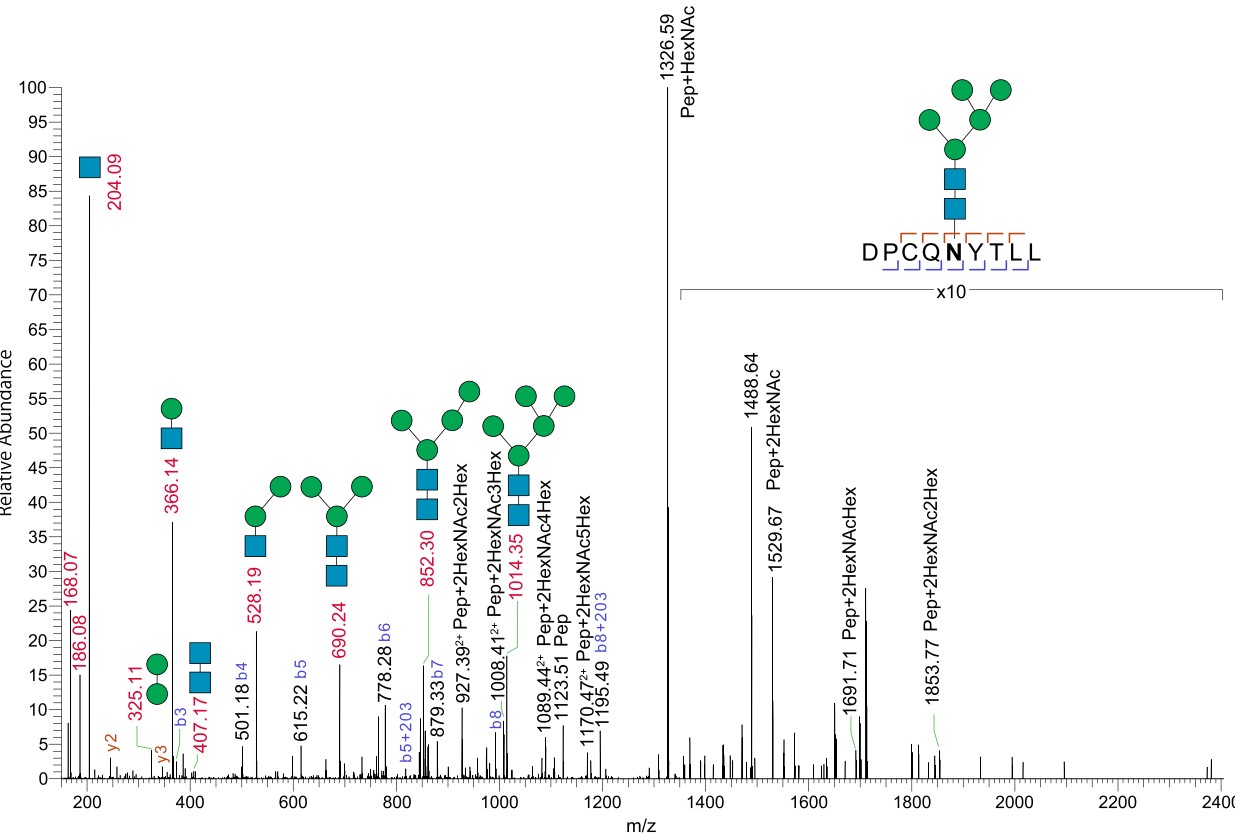

**Extended Data Fig. 8 | Mass spectrometric analysis of GP2 glycopeptides detects the oligomannose-5 structure attached to N65.** Supporting MS2 spectrum of precursor m/z 1170.46, [61]DPCQNYTLL[69], carrying oligomannose-5 (HexNAc2Hex5). Prepared by Asp-N digestion of the GP2 branch purified from HEK293T cells. N-glycan structures are depicted following the Consortium for Functional Glycomics (CFG) notation: HexNAc, N-acetylglucosamine (blue square); Hex, mannose (green circle). The cysteine residue is carbamidomethylated. Detected peptide-backbone fragment ions are presented in the peptide sequence. Interestingly, complex-type carbohydrate structures were also found to be attached to N65. This is consistent with the observation that, although UMOD N275 and GP2 N65 are both located in the groove between the β-hairpin and the D10C domain of the respective decoy modules, N65 is relatively more exposed than N275 in the structure (Extended Data Fig. 7a), making the N65 glycan chains more susceptible to modification.

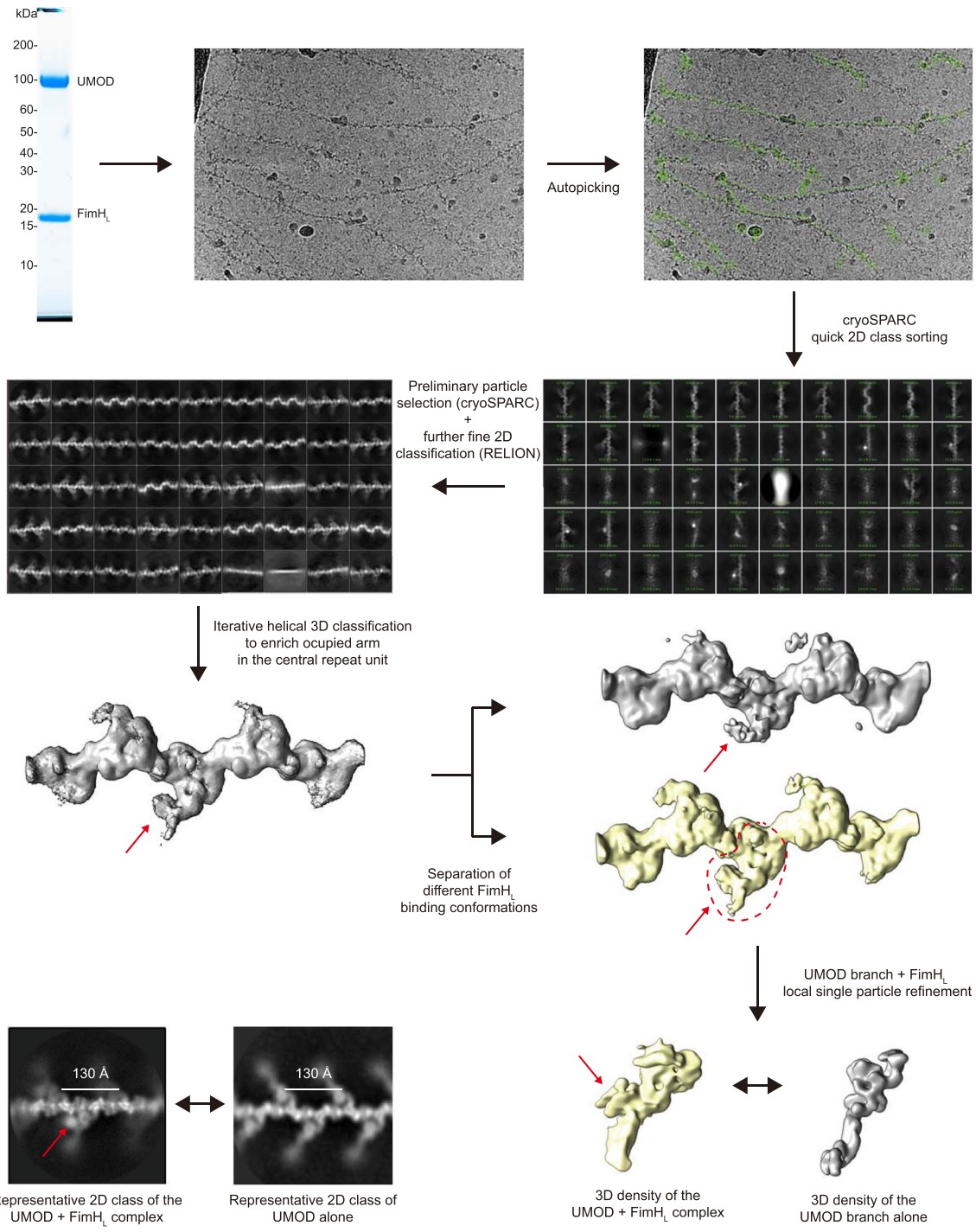

**Extended Data Fig. 9 | 3D reconstruction of the UMOD branch/FimH_L complex.** Identification, isolation and local refinement of a single UMOD branch unit bound to one copy of FimH_L. After incubation with an excess concentration of FimH_L, UMOD filaments were subjected to cryo-EM analysis. Following filament autopicking by an in-house script, highly heterogenous filament segments were sorted by performing cryoSPARC 2D class runs, after binning. Segment coordinates from good 2D classes were then extracted and re-imported into RELION. After iterative 3D classification with and without applying helical symmetry, the segments with higher FimH_L occupancy were selected and grouped into different sub-classes. Segments representing a single branch unit of the best UMOD/FimH_L sub-class were extracted and used for 3D reconstruction of the density of UMOD bound to FimH_L. In the bottom left panel, the extra density of FimH_L in the UMOD branch/FimH_L complex could be identified in the 2D class images. Red arrows point to the location of FimH_L.

# Reporting Summary

## Statistics

For all statistical analyses, confirm that the following items are present in the figure legend, table legend, main text, or Methods section.

| n/a | Confirmed | |
|---|---|---|
| ☐ | ☒ | The exact sample size (*n*) for each experimental group/condition, given as a discrete number and unit of measurement |
| ☐ | ☒ | A statement on whether measurements were taken from distinct samples or whether the same sample was measured repeatedly |
| ☒ | ☐ | The statistical test(s) used AND whether they are one- or two-sided *Only common tests should be described solely by name; describe more complex techniques in the Methods section.* |
| ☒ | ☐ | A description of all covariates tested |
| ☒ | ☐ | A description of any assumptions or corrections, such as tests of normality and adjustment for multiple comparisons |
| ☒ | ☐ | A full description of the statistical parameters including central tendency (e.g. means) or other basic estimates (e.g. regression coefficient) AND variation (e.g. standard deviation) or associated estimates of uncertainty (e.g. confidence intervals) |
| ☒ | ☐ | For null hypothesis testing, the test statistic (e.g. *F*, *t*, *r*) with confidence intervals, effect sizes, degrees of freedom and *P* value noted *Give P values as exact values whenever suitable.* |
| ☒ | ☐ | For Bayesian analysis, information on the choice of priors and Markov chain Monte Carlo settings |
| ☒ | ☐ | For hierarchical and complex designs, identification of the appropriate level for tests and full reporting of outcomes |
| ☒ | ☐ | Estimates of effect sizes (e.g. Cohen's *d*, Pearson's *r*), indicating how they were calculated |

*Our web collection on statistics for biologists contains articles on many of the points above.*

## Software and code

Policy information about availability of computer code

| Data collection | EPU 2.11.1, MXCuBE3 3.0 |
|---|---|
| Data analysis | AlphaFold Monomer 2.0; AS2TS server 09/2019; ByonicTM 3.4.0; ConSurf 2016; Coot 0.8.9.2-0.9.6; Cryo-EM-filament-picker 1.0; cryoSPARC 3.2; CTFFIND 4.1.14; Dali 5; ESPript 3.0; ISOLDE 1.2; MAFFT 7; MODELLER 10.1; MolProbity 4.5.1; MView 1.68; Namdinator 2.0; Phaser 2.8.3; PHENIX (phenix.autobuild, phenix.refine, phenix.table_one) 1.19.2_4158, dev_4282; Privateer MKIII-MKIV; PyMOL 2.4.2; RELION 3.0.8; Scipion 3.0.9; STRIDE 1.0; TEMPy:DiffMap 2; UCSF Chimera 1.11-1.15; UCSF ChimeraX 1.1-1.2; XDS Feb 5, 2021 BUILT=20210322; YASARA Structure 21.4.22 |

For manuscripts utilizing custom algorithms or software that are central to the research but not yet described in published literature, software must be made available to editors and reviewers. We strongly encourage code deposition in a community repository (e.g. GitHub). See the Nature Portfolio guidelines for submitting code & software for further information.

## Data

Policy information about availability of data

All manuscripts must include a data availability statement. This statement should provide the following information, where applicable:
- Accession codes, unique identifiers, or web links for publicly available datasets
- A description of any restrictions on data availability
- For clinical datasets or third party data, please ensure that the statement adheres to our policy

The UniProt (https://www.uniprot.org/) IDs for hGP2 and hUMOD are P55259 and P07911, respectively; the IDs of other sequences reported in the alignment of Extended Data Fig. 1b are Q9D733 (mGP2), Q91X17 (mUMOD), Q8WWZ8 (hLZP), Q8R4V5 (mLZP), Q8N2E2 (hVWDE) and Q6DFV8 (mVWDE). The Electron Microscopy Data Bank (EMDB; https://www.ebi.ac.uk/emdb/) ID of the UMOD filament map used for assembling the composite map shown in this work is

EMD-10553; the UMOD filament core and FimHL/trimannose coordinates used as starting models can be retrieved from the Protein Data Bank (PDB; http://www.rcsb.org) with IDs 6TQK and 6GTW, respectively.

Structure factors and atomic models for the P1, P212121 and C2 crystal forms of the GP2 decoy domain have been deposited in the PDB with accession codes 7P6R, 7P6S and 7P6T, respectively. Cryo-EM density maps of full-length UMOD and the UMOD branch + EGF IV/FimHL complex have been deposited in the EMDB with accession codes EMD-13378 and EMD-13794, respectively; the corresponding coordinates have been deposited in the PDB with accession codes 7PFP and 7Q3N.

# Field-specific reporting

Please select the one below that is the best fit for your research. If you are not sure, read the appropriate sections before making your selection.

☒ Life sciences          ☐ Behavioural & social sciences          ☐ Ecological, evolutionary & environmental sciences

For a reference copy of the document with all sections, see nature.com/documents/nr-reporting-summary-flat.pdf

# Life sciences study design

All studies must disclose on these points even when the disclosure is negative.

| Sample size | No statistical methods were used to predetermine sample size.<br>For structure determination of the GP2 decoy module by X-ray crystallography, we measured diffraction from specimens that belonged to three different crystal forms (P1, P2(1)2(1)2(1) and C2). Using crystals harvested from multiple crystallization drops, we screened 119 samples and collected 13 P1, 15 P2(1)2(1)2(1) and 3 C2 datasets. The datasets belonging to each space group were then ranked by resolution and quality (based on the statistical indicators reported in Supplementary Table 1), and the best ones (which were processed to resolutions of 1.9 Å, 1.35 Å and 1.4 Å, respectively; Supplementary Table 1) were used for structure solution by molecular replacement and refinement.<br>For cryo-EM analysis of full-length UMOD and the UMOD branch/FimH(L) complex, we screened >20 grids of each sample at 0.8-1.8 mg mL-1 concentrations. The datasets used for structure determination consisted of 2,300 and 13,616 raw micrographs, respectively, from which 412,322 and 3,767,790 filaments were picked and used for 2D classification. The number of particles used in the final reconstructions was 288,403 (UMOD filament core + D10C domain), 114,206 (UMOD branch) and 225,819 (UMOD branch/FimH(L)) (Supplementary Table 3). This was sufficient to assemble a composite map of full-length UMOD with a nominal resolution of 6.1 Å, and to obtain a map of the UMOD branch/FimH(L) complex with a nominal resolution of 7.4 Å.<br>For biochemical experiments, we used amounts and concentrations of proteins that provided sufficient signal-to-noise ratios to obtain unambiguous results, based on previous knowledge of the corresponding experimental setups. |
|---|---|
| Data exclusions | As in the case of all single-crystal X-ray diffraction experiments, two high resolution choices were made for each dataset that could have at least potentially excluded part of the weakest reflections: first, a crystal-to-detector distance was chosen, based on an initial resolution estimate made by the beamline data collection/processing software; second, a more accurate high-resolution cutoff was chosen, based on the mean I/σI and CC(1/2) values obtained upon manual processing of the datasets. The latter choice was made following the established criteria described in PMIDs 23793146 and 26209821 (Methods-associated references 29 and 30).<br>Processing of the cryo-EM data for full-length UMOD has already been described in PMID 33196145 (reference 6 of the manuscript). For determining the structure of the UMOD branch/FimH(L) complex by cryo-EM, we only processed micrographs with an estimated resolution better than 8 A. As also detailed in the Methods, subsequent particle exclusions were performed at three different stages: (1) starting from a total of 13,616 raw micrographs, 3,767,790 helical segments were auto-picked and extracted on the basis of motion correction and CTF estimation; (2) based on 2D classification quality evaluated with cryoSPARC, a subset of 1,139,808 particles was then selected for further processing; and (3) because FimH(L) occupancy varied among filaments, segments with higher FimH(L) occupancy were selected during iterative RELION 3D classification runs, resulting in 225,819 homogeneous particles that were subjected to auto-refinement and postprocessing.<br>Finally, no data was excluded in conjunction with the biochemical experiments described in this manuscript. |
| Replication | Although the structures of the three crystal forms of the GP2 decoy module were obtained from diffraction data collected from single crystals (as commonly done in X-ray crystallography), as detailed in the section "Sample size" several specimens were screened and measured for each of them. For each crystal form, all of these samples were consistent in terms of morphology, space group and unit cell dimensions. Most importantly, the structures of the three different crystal forms of the protein are essentially equivalent (average Cα RMSD 0.6 A).<br>Cryo-EM single particle analysis averages independent particle observations, and – as reported in Supplementary Table 3 – 288,403 and 225,819 particles were averaged to yield the final 3D reconstructions of full-length UMOD and the UMOD branch/FimH(L) complex, respectively.<br>Biochemical experiments were successfully reproduced as detailed in the respective figure legends. Specifically, n=3 for the experiments shown in Fig. 1b, Fig. 2c, Extended Data Fig. 2b (GP2 decoy module, UMOD decoy module), Extended Data Fig. 2c and n=2 for the experiments of Extended Data Fig. 2a, Extended Data Fig. 2b (GP2 decoy module/FimH(L), UMOD decoy module/FimH(L)) and Extended Data Fig. 7b (GP2 decoy module/FimH(L), GP2 decoy module N65A/FimH(L)). |
| Randomization | X-ray crystallography: random assignment of reflections to working or free sets was automatically performed by PHENIX (P1 data) or XDS (P2(1)2(1)2(1) and C2 data).<br>Cryo-EM: The vitrified UMOD filaments (free or bound to FimH) used for structure determination by cryo-EM adopt random orientations on the XY plane of the EM grids, although – as previously described in PMID 33196145/reference 6 of the manuscript – they are significantly less randomly distributed along Z due to the fact that they tend to lie flat on the grids themselves. Assignment of particles into random half datasets was automatically performed by RELION during 3D reconstruction.<br>Biochemical experiments: these experiments did not involve or require randomization. |
| Blinding | Blinding was not applicable to the type of data that was analyzed in this study. In particular, knowledge of the identity of the molecules under investigation was required to express them, purify them and determine their structure, because the success of all these procedures depends on information (primary sequence, post-translational modifications etc.) that is specific to each experimental sample. |

# Reporting for specific materials, systems and methods

We require information from authors about some types of materials, experimental systems and methods used in many studies. Here, indicate whether each material, system or method listed is relevant to your study. If you are not sure if a list item applies to your research, read the appropriate section before selecting a response.

## Materials & experimental systems

| n/a | Involved in the study |
|-----|----------------------|
| ☐ | ☒ Antibodies |
| ☐ | ☒ Eukaryotic cell lines |
| ☒ | ☐ Palaeontology and archaeology |
| ☒ | ☐ Animals and other organisms |
| ☐ | ☒ Human research participants |
| ☒ | ☐ Clinical data |
| ☒ | ☐ Dual use research of concern |

## Methods

| n/a | Involved in the study |
|-----|----------------------|
| ☒ | ☐ ChIP-seq |
| ☒ | ☐ Flow cytometry |
| ☒ | ☐ MRI-based neuroimaging |

## Antibodies

| | |
|---|---|
| Antibodies used | Primary antibody: Penta-His Antibody, BSA-free (QIAGEN, Cat. No. 34660, Lot 157046697).<br>Secondary antibody: Goat anti-Mouse IgG Fc Secondary Antibody, HRP (Invitrogen, Cat. No. A16084, Lot 62-47-012318). |
| Validation | The QIAGEN Penta-His Antibody is an anti-(H)5 mouse monoclonal for the "highly specific detection of C-terminal, N-terminal and internal His tags". As described on the product's web page (https://www.qiagen.com/se/products/discovery-and-translational-research/protein-purification/tagged-protein-expression-purification-detection/anti-his-antibodies-bsa-free/?catno=34660) and in the QIAexpress® Detection and Assay Handbook (Fourth Edition/July 2015) that can be downloaded from the same URL, this antibody recognizes its epitope with nanomolar affinity, can detect ~50 pg protein in Western blots (using a chemiluminescent substrate) and has been validated against many different proteins. We have abundantly used it in our previous work (see for example PMID 26850170), and repeatedly validated it by also using it to probe, as negative controls, conditioned media samples from cells that that do not express His-tagged protein. |

## Eukaryotic cell lines

Policy information about cell lines

| | |
|---|---|
| Cell line source(s) | HEK293T: laboratory of Prof. A. Radu Aricescu (University of Oxford, UK; now at the MRC Laboratory of Molecular Biology, Cambridge, UK) (PMID 3031469); the commercial source for this cell line was ATCC cat. no. CRL-3216, RRID CVCL_0063.<br>Expi293F GnTI-: Thermo Fisher Scientific cat. no. A39240. |
| Authentication | Cell line authentication was performed by the commercial sources described above, which guarantee their authenticity; no additional authentication was performed by either Prof. Aricescu or our laboratory. However, even though we did not verify cell line identities genetically, the results reported in this manuscript and other work in the laboratory showed that the type of glycosylation of the recombinant proteins expressed in these cell lines was consistent with their expected genetic background. Namely, enzymatic deglycosylation and/or mass spectrometric analysis showed that the glycans attached to recombinant proteins expressed in HEK293T were mostly complex-type (except in notable cases such as UMOD N275 (Fig. 2c) and GP2 N65 (Extended Data Fig. 8)), whereas those attached to proteins expressed in Expi293F GnTI- cells were high-mannose-type. |
| Mycoplasma contamination | Each cell line was tested for mycoplasma contamination by the respective source. We confirmed that the HEK293T cell line was mycoplasma-free by using a PCR Mycoplasma Test Kit II (Applichem cat. no. A8994). |
| Commonly misidentified lines<br>(See ICLAC register) | No commonly misidentified cell lines were used. |

## Human research participants

Policy information about studies involving human research participants

| | |
|---|---|
| Population characteristics | The research participant is a healthy male, who was 49 year old at the time of sample collection. |
| Recruitment | The participant is one of the authors of the manuscript (L.J.), who received no compensation. |
| Ethics oversight | No ethical approval was deemed necessary by the participant's department (Karolinska Institutet, Department of Biosciences and Nutrition), as he used his own urine. |

Note that full information on the approval of the study protocol must also be provided in the manuscript.

