## [Peer Review File · Nature Structural & Molecular Biology]

Peer Review Information

Journal: Structural and Molecular Biology

Manuscript Title: Structure of the decoy module of human glycoprotein 2 and uromodulin and its interaction with bacterial adhesin FimH

Corresponding author name(s): Professor Luca Jovine

Reviewer Comments & Decisions:

Decision Letter, initial version:
--

14th Oct 2021

Dear Dr. Jovine,

Thank you again for submitting your manuscript "Structure of the bacteria-capturing decoy module of human glycoprotein 2 and uromodulin". I apologize for the delay in responding, which resulted from the difficulty in obtaining suitable referee reports. Nevertheless, we now have comments (below) from the 2 reviewers who evaluated your paper. In light of those reports, we remain interested in your study and would like to see your response to the comments of the referees, in the form of a revised manuscript.

You will be pleased to see that both reviewers are positive about the potential interest of the findings and quality of the data. Each also notes issues that should be addressed by revisions. Reviewer #1, an expert in structural biology of glycoproteins, finds that the experimental approach should be described in more detail, that aspects of data interpretation be clarified, and also offers suggestions to improve data presentation. Reviewer #2, an expert in protein structure modeling, asks for further clarification of how homology modeling was done, and additional information regarding the AF2 outputs. Editorially, we agree that these suggestions would strengthen the work, and ask that they be included in a revised manuscript.

Please be sure to address/respond to all concerns of the referees in full in a point-by-point response and highlight all changes in the revised manuscript text file. If you have comments that are intended for editors only, please include those in a separate cover letter.

We expect to see your revised manuscript within 3 weeks. If you cannot send it within this time,

please contact us to discuss an extension; we would still consider your revision, provided that no similar work has been accepted for publication at NSMB or published elsewhere.

Reporting Summary:

Please note that all key data shown in the main figures as cropped gels or blots should be presented in uncropped form, with molecular weight markers. These data can be aggregated into a single supplementary figure item. While these data can be displayed in a relatively informal style, they must refer back to the relevant figures. These data should be submitted with the final revision, as source data, prior to acceptance, but you may want to start putting it together at this point.

Data availability: this journal strongly supports public availability of data. All data used in accepted papers should be available via a public data repository, or alternatively, as Supplementary Information. If data can only be shared on request, please explain why in your Data Availability Statement, and also in the correspondence with your editor. Please note that for some data types, deposition in a public repository is mandatory - more information on our data deposition policies and available repositories can be found below:

<https://www.nature.com/nature-research/editorial-policies/reporting-standards#availability-of-data>

We require deposition of coordinates (and, in the case of crystal structures, structure factors) into the Protein Data Bank with the designation of immediate release upon publication (HPUB). Electron microscopy-derived density maps and coordinate data must be deposited in EMDB and released upon publication. Dataset accession numbers must be supplied with the final accepted manuscript and appropriate release dates must be indicated at the galley proof stage.

[REDACTED]

With kind regards,

Beth

Beth Moorefield, Ph.D.
Senior Editor
Nature Structural & Molecular Biology

Reviewers' Comments:

Reviewer #1:

Remarks to the Author:

Last year, the groups of Luca Jovine and Rudi Glockshuber reported cryo-EM structures of the uromodulin filament that acts as a bacteria-binding decoy in the urinary tract. The ZP domain core of the filaments could be modelled at atomic resolution, but the lack of structural information about the D8C domain precluded a modelling of the branches. The D8C domain is functionally important because it contains the high-mannose glycan that mediates binding to the bacterial FimH protein.

In the present manuscript, Stsiapaneva et al. report: 1) the crystal structure of the D8C domain of glycoprotein 2 (60% identity to uromodulin) – now renamed D10C as it was found to contain another pair of cysteines at the N-terminus; 2) a cryo-EM structure of the uromodulin filament with FimH bound to D10C. The latter structure at 7.4 Å resolution reveals much more detail than the previous structure from the Glockshuber lab obtained by tomography (Weiss et al., Science, 2020). In particular, the new cryo-EM structure clearly shows that there is one FimH protein bound to each branch, and not two as suggested by the Glockshuber study. The new structure further shows that the high-mannose glycan is located in a groove, such that it is protected from enzymatic digestion. Finally, the study is interesting from a technical perspective, as the crystal structure was solved using an AlphaFold model (although I suspect that there will be many similar cases forthcoming soon).

Overall, these are important findings that complete the picture of the uromodulin-FimH assembly. The novelty and narrow focus of the study make it suitable for a Brief Communication.

The technical quality of the structure determinations is high, and the results are well presented (but see my comments 4 and 5).

There are a number of points that the authors should address before publication:

1. Why did the authors use a periplasmic extract instead of purified FimH? Panel c of Figure S2 suggests that purification should be feasible.
2. Please provide experimental detail of how the uromodulin-FimH filaments were prepared. What was the FimH:uromodulin stoichiometry? Is it possible that a second FimH binding site was missed because of insufficient FimH?
3. The results imply that Asn65 in glycoprotein 2 carries a high-mannose glycan. Could this be confirmed by mass spectrometry or EndoH digestion (as in Figure 2c)?
4. The binding experiments in Figure S2 are not presented ideally. The vertical arrangement of panel a is confusing. I am not sure the strips for UMOD and GP2 are needed, given that the electrophoretic profile of the proteins is also shown in the insets of panel b. The chromatographic traces of the isolated decoy modules should be shown for comparison (panel b, top and middle chromatograms).
5. Panels c-g in Figure 1 cannot be understood without reference to Table S2. Information on the relevant disease mutations should be added to the figures or the legend.

Reviewer #2:

Remarks to the Author:

The authors present an interesting study, inferring molecular details of filaments with an unusual bacterial decoy function. The overall approach is sound and I couldn't see any holes in the logic. It mixes cryo-EM, crystal, biochemical and bioinformatic approaches in a pleasing way, and includes a timely use of AlphaFold 2, both for Molecular Replacement and for cryoEM map fitting. The description is generally admirably clear and concise and the figures excellent. I have only a few comments

- Supp Table 2 focusing on pathogenic mutations is generally well-grounded (although we hardly need the structure to predict that disrupting (likely) disulfide bonds will be detrimental). A possible exception is P236L where the substitution does not change size and physicochemical characteristics dramatically. Is the Pro in an unusual area of the Ramachandran plot? Do methods like FoldX predict a significant loss of stability?

- the homology modelling on p.17 has some issues. Although sequence identity is high it could have been done more carefully: different positions of the 1-residue deletion could have been trialled (the alignment does not seem completely unambiguous here); more advanced model quality measurements than DOPE could have been used; it is well known that standard MD (i.e. without constraints and using standard force fields) is more likely to degrade a model than refine it. However, since the results were only used as guidance I think these issues can be addressed simply by removing the word 'refinement' from 'refined by molecular dynamics in YASARA' and replacing it with a phrase indicating exploration of different conformations.

- especially given the excitement around AF2 it would be interesting to see somewhere a comment as to whether the structure would have been determinable if the crystal structure had yielded to experimental phasing, or if AF2 had an essential second participation in predicting inter-domain orientations when interpreting the cryoEM maps

- again given the AF2 excitement I would be interested to know -- if the PAE predicted that the hairpin and the D10C domain relative orientation was not guaranteed to be correct
- if the Cys residues were modelled in such a way that software like PyMOL directly inferred disulphide bonds. As I understand it, disulphide bonds are not explicitly featured in AF2 outputs.
- minor point. In ED fig 4 upper panel, the blue domain linker between hairpin and D10C is practically invisible. Might a different superposition eg on D10C domain alone make it visible?

Author Rebuttal to Initial comments

Response to Reviewer #1:

Remarks to the Author:

Last year, the groups of Luca Jovine and Rudi Glockshuber reported cryo-EM structures of the uromodulin filament that acts as a bacteria-binding decoy in the urinary tract. The ZP domain core of the filaments could be modelled at atomic resolution, but the lack of structural information about the D8C domain precluded a modelling of the branches. The D8C domain is functionally important because it contains the high-mannose glycan that mediates binding to the bacterial FimH protein.

In the present manuscript, Stsiapaneva et al. report: 1) the crystal structure of the D8C domain of glycoprotein 2 (60% identity to uromodulin) – now renamed D10C as it was found to contain another pair of cysteines at the N-terminus; 2) a cryo-EM structure of the uromodulin filament with FimH bound to D10C. The latter structure at 7.4 Å resolution reveals much more detail than the previous structure from the Glockshuber lab obtained by tomography (Weiss et al., Science, 2020). In particular, the new cryo-EM structure clearly shows that there is one FimH protein bound to each branch, and not two as suggested by the Glockshuber study. The new structure further shows that the high-mannose glycan is located in a groove, such that it is protected from enzymatic digestion. Finally, the study is interesting from a technical perspective, as the crystal structure was solved using an AlphaFold model (although I suspect that there will be many similar cases forthcoming soon).

Overall, these are important findings that complete the picture of the uromodulin-FimH assembly. The novelty and narrow focus of the study make it suitable for a Brief Communication.

The technical quality of the structure determinations is high, and the results are well presented (but see my comments 4 and 5).

There are a number of points that the authors should address before publication:

We thank the Reviewer for summarizing our findings in the context of previously available information, and very much appreciate their recognition of the novelty and interest of the work described in this manuscript.

Point 1

Why did the authors use a periplasmic extract instead of purified FimH_L? Panel c of Figure S2 suggests that purification should be feasible.

The reason for using a periplasmic extract instead of purified FimH_L was to show that not only the GP2 and UMOD decoy modules are able to bind the lectin, but they can also selectively fish it out from a complex protein mixture.

To verify the identity of the ~15 kDa bands fished out by the GP2 and UMOD decoy modules we have now analyzed them by mass spectrometry:

As can be seen from the MASCOT search summaries attached below, MS/MS analysis confirmed that both bands indeed correspond to FimH_L.

This information has now been added to the legend of Extended Data Fig. 2b (page 38, lines 659-663): *"In both cases, reducing SDS-PAGE of peak fractions and tandem mass spectrometry (MS/MS) of the corresponding ~15 kDa bands show the presence of complexes between the decoy modules and the bacterial adhesin, indicating that the former are able to selectively recognize the latter among the pool of periplasmic proteins."*

F047730

Header																																													
<pre> Search title ~ Max number of peaks in MS/MS spectra - 900~ Timestamp 2021-10-21T04:29:27Z User Protlioklied Email akos.vegvani@ki.se Report URI http://10.237.65.246/mascot/cgi/master_results.pl?file=~/data/20211021/F047730.dat Peak list data path Peak list format Mascot generic Search type MIS Mascot version 2.5.1 Database CP_EColi Fasta file CP_EColi_20170609.fasta Total sequences 4316 Total residues 1361389 Sequences after taxonomy filter 4316 Number of queries 11253 </pre>																																													
Decoy																																													
<pre> Number of matches above identity 1 133 Number of matches above identity 1 1 </pre>																																													
Variable modifications																																													
  Identifier Name Delta Neutral loss(es)   1 Carbamidomethyl (C) 57,02146    2 Deamidated (NQ) 0,984016    3 Oxidation (M) 15,99492   	Identifier	Name	Delta	Neutral loss(es)	1	Carbamidomethyl (C)	57,02146		2	Deamidated (NQ)	0,984016		3	Oxidation (M)	15,99492		0 63,998285																												
Identifier	Name	Delta	Neutral loss(es)																																										
1	Carbamidomethyl (C)	57,02146																																											
2	Deamidated (NQ)	0,984016																																											
3	Oxidation (M)	15,99492																																											
Search Parameters																																													
<pre> Taxonomy filter All entries Enzyme Trypsin Maximum Missed Cleavages 2 Fixed modifications Variable modifications Carbamidomethyl (C),Deamidated (NQ),Oxidation (M) Peptide Mass Tolerance 10 Peptide Mass Tolerance Units ppm Fragment Mass Tolerance 0,02 Fragment Mass Tolerance Units Da Mass values Monoisotopic Instrument type Default Decoy database also searched 1 </pre>																																													
Format parameters																																													
<pre> Significance threshold 0,05 Max. number of hits 0 Use MudPIT protein scoring 1 Ions score cut-off 0 Include same-set proteins 1 Include sub-set proteins 1 Include unassigned 0 Require bold red 0 Use homology threshold 1 Group protein families 1 Re-score using Percolator 1 Show duplicate peptides 0 </pre>																																													
Protein hits																																													
   prot_hit_num prot_family_member prot_acc prot_desc prot_score prot_mass prot_matches prot_matches prot_sequences prot_sequences prot_cover     1 1 P08191 Type 1 fimbriae D-mannose specific adhesin OS=Escherichia coli (strain K12) GN=fimH PE=1 SV=2 2265 31454 188 131 5 5 22   2 1 P46889 DNA translocase FtsK OS=Escherichia coli (strain K12) GN=ftsK PE=1 SV=2 19 146572 1 1 1 1 0,6   3 1 P32715 Multidrug resistance protein MDR OS=Escherichia coli (strain K12) GN=mdrO PE=1 SV=2 14 76101 1 1 1 1 1,5   	prot_hit_num	prot_family_member	prot_acc	prot_desc	prot_score	prot_mass	prot_matches	prot_matches	prot_sequences	prot_sequences	prot_cover	1	1	P08191	Type 1 fimbriae D-mannose specific adhesin OS=Escherichia coli (strain K12) GN=fimH PE=1 SV=2	2265	31454	188	131	5	5	22	2	1	P46889	DNA translocase FtsK OS=Escherichia coli (strain K12) GN=ftsK PE=1 SV=2	19	146572	1	1	1	1	0,6	3	1	P32715	Multidrug resistance protein MDR OS=Escherichia coli (strain K12) GN=mdrO PE=1 SV=2	14	76101	1	1	1	1	1,5	
prot_hit_num	prot_family_member	prot_acc	prot_desc	prot_score	prot_mass	prot_matches	prot_matches	prot_sequences	prot_sequences	prot_cover																																			
1	1	P08191	Type 1 fimbriae D-mannose specific adhesin OS=Escherichia coli (strain K12) GN=fimH PE=1 SV=2	2265	31454	188	131	5	5	22																																			
2	1	P46889	DNA translocase FtsK OS=Escherichia coli (strain K12) GN=ftsK PE=1 SV=2	19	146572	1	1	1	1	0,6																																			
3	1	P32715	Multidrug resistance protein MDR OS=Escherichia coli (strain K12) GN=mdrO PE=1 SV=2	14	76101	1	1	1	1	1,5																																			

F047731

Header									
Search title	~ Max number of peaks in MS/MS spectra - 900~								
Timestamp	2021-10-21T04:30:20Z								
User	ProtBioMed								
Email	alios.segami@ki.se								
Report URI	http://10.237.65.246/mascot/cgi/master_results.pl?file=./data/20211021/F047731.dat								
Peak list data path	\\user.ki.se\labs\Proteomics\PROTEOMICS CORE FACILITIES\Proteomics Biomedicum\Q Ex HF 2\2021\20211020_nU3_E5903_2104196_PB_122\20211020_HF2_02_SN_2.mgf								
Peak list format	Mascot generic								
Search type	MIS								
Mascot version	2.5.1								
Database	CP_EColi								
Fasta file	CP_EColi_20170609.fasta								
Total sequences	4316								
Total residues	1361389								
Sequences after taxonomy filter	4316								
Number of queries	11305								
Decoy									
Number of matches above identity %	146								
Number of matches above identity %	0								
Variable modifications									
Identifier	Name Delta Neutral loss(es)								
	1 Carbamidomethyl (C) 57.021464								
	2 Deamidated (NQ) 0.984016								
	3 Oxidation (M) 15.994915								
	0 63.998285								
Search Parameters									
Taxonomy filter	All entries								
Enzyme	Trypsin								
Maximum Missed Cleavages	2								
Fixed modifications									
Variable modifications	Carbamidomethyl (C), Deamidated (NQ), Oxidation (M)								
Peptide Mass Tolerance	10								
Peptide Mass Tolerance Units	ppm								
Fragment Mass Tolerance	0.02								
Fragment Mass Tolerance Units	Da								
Mass values	Monoisotopic								
Instrument type	Default								
Decoy database also searched	1								
Format parameters									
Significance threshold	0.05								
Max. number of hits	0								
Use MudPIT protein scoring	1								
Ions score cut-off	0								
Include same-set proteins	1								
Include sub-set proteins	1								
Include unassigned	0								
Require bold red	0								
Use homology threshold	1								
Group protein families	1								
Re-score using Percolator	1								
Show duplicate peptides	0								
Protein hits									
prot_hit_num	prot_family_member prot_acc prot_desc prot_score prot_mass prot_matche prot_matche prot_sequens prot_sequens prot_cover								
1	1 P08191	Type 1 fimbria D-mannose specific adhesin O5=Escherichia coli (strain K12) GN=FimH PE=1 SV=2	3155	31454	176	145	6	6	24
2	1 P45762	Putative type II secretion system protein K O5=Escherichia coli (strain K12) GN=sgk PE=2 SV=1	21	37623	1	1	1	1	11.6

Point 2

Please provide experimental detail of how the uromodulin-FimH filaments were prepared. What was the FimH:uromodulin stoichiometry? Is it possible that a second FimH binding site was missed because of insufficient FimH?

The FimH_L construct used to obtain the cryo-EM reconstruction described in the manuscript was from uropathogenic *E. coli* strain UTI89 (more specifically, the A27V variant that - in combination with V163A at the beginning of the FimH pilin domain - was previously shown to bind with high affinity to different mannosylated proteins, including UMOD (Kalas *et al.* (2017), cited in the revised manuscript as reference 23)). This information has now been added to the revised manuscript in the main text (pages 4-5, lines 86-87: “we reconstituted *in vitro* the complex between UMOD and FimH_L from uropathogenic *E. coli* (UPEC) UTI89”), the legend of Fig. 2d (page 11, lines 167-168: “Recognition of the D10C N275 glycan by the lectin domain of fimbrial adhesin FimH from UPEC UTI89.”) and the “Protein expression and purification” section of the Methods (page 13, lines 202-203: “His-tagged FimH_L A27V from UPEC strain UTI89²³”).

As described in the same section of the Methods (page 13, lines 207-208), we reconstituted the UMOD:FimH_L complex using a 1:3 molar ratio, which is close to the 1:4 ratio

reported for the cryo-ET studies by Weiss *et al.* (2020). Despite the fact that we observed significant heterogeneity in the sample (as written in the legend of Extended Data Fig. 9), after iterative 2D classification of all UMOD segments we were only able to identify 1:1 UMOD:FimH_L complexes. The more helical segments we included in the analysis, the more apparent the binding ratio:

Although our 2D classes are not consistent with a 1:2 complex, this does not formally exclude the possibility that such a species may have been present in a minority of cases. Whereas we can of course only speak for our own data, one aspect that should be taken into account in this regard is that, in subtomogram averaging at low resolution, 1:1 complexes images would be averaged out with 1:2 complexes.

In relation to this point, it should also be considered that previous work by the Glockshuber group showed “that a single N-glycan can bind up to three molecules of FimH, albeit with negative cooperativity, so that a molar excess of accessible N-glycans over FimH on the cell surface favors monovalent FimH binding”; moreover, as also depicted in the graphical abstract of the same publication, 1:1 binding is expected to be predominant under physiological flow conditions (reference 47: Sauer *et al. J. Am. Chem. Soc.* **141**, 936–944 (2019)). Although the static conditions of cryo-EM/ET experiments clearly do not rule out multivalent binding when there is a molar excess of FimH over UMOD, the 1:1 stoichiometry visualised by our cryo-EM reconstruction is thus more likely to recapitulate the UMOD/FimH interaction that takes place in the urine.

Point 3

The results imply that Asn65 in glycoprotein 2 carries a high-mannose glycan. Could this be confirmed by mass spectrometry or EndoH digestion (as in Figure 2c)?

Unlike the UMOD decoy module construct used for the deglycosylation experiment shown in Fig. 2c, which carries only one glycan (N232) in addition to the one attached to N275, the GP2 branch/decoy module has 4 N-glycosylation sites (N65, N88, N122 and N134). As visible in the SDS-PAGE gels shown in Extended Data Figs. 2b and 7b, when the GP2 construct is expressed in HEK293T cells this makes the protein highly heterogeneous. Because this precluded the possibility of getting a clear Endo-H digestion pattern like the one obtained for UMOD, we followed the suggestion of the Reviewer and collaborated with Dr. Nao Yamakawa, an expert of protein glycosylation analysis by mass spectrometry who has now been also added as a co-author of the manuscript, to investigate whether GP2 N65 indeed carries a high-mannose glycan. As now reported in the main text and shown in a new figure (page 4, lines 81-83: *“and mass spectrometric analysis of the glycans attached to N65 detects the HexNAc2Hex5 oligomannose structure (Extended Data Fig. 8)”*), LC-MS/MS analysis of the N65 glycopeptides (described in a newly added Methods section *“Site specific N-glycosylation analysis by liquid chromatography–tandem mass spectrometry (LC-MS/MS)”*, pages 22-23) confirmed our previous mutagenesis results by detecting the high-mannose structure.

Interestingly, this analysis showed that not only N65 is highly glycosylated, but also that its glycosylation is heterogeneous, so that only part of the glycans attached to N65 are high-mannose type. As we also reported in the legend of the new figure, *“This is consistent with the observation that, although UMOD N275 and GP2 N65 are both located in the groove between the β -hairpin and the D10C domain of the respective decoy modules, N65 is relatively more exposed than N275 in the structure (Extended Data Fig. 7a), making the N65 glycan chains more susceptible to modification.”*

By combining this new mass spectrometry data to our previous *in vitro* binding and mutagenesis results, we conclude that a significant proportion of GP2 molecules carry a high-mannose glycan attached to N65, and that - as in the case of the UMOD N275 sugar - this glycan allows GP2 to capture FimH (for consistency with UMOD N275's, the GP2 N65 glycan in Extended Data Fig. 1a,b is now also colored cyan). The additional analysis mentioned by the Reviewer clearly made this important point more conclusive, and we are thus grateful for their constructive suggestion.

Point 4

The binding experiments in Figure S2 are not presented ideally. The vertical arrangement of panel a is confusing. I am not sure the strips for UMOD and GP2 are needed, given that the electrophoretic profile of the proteins is also shown in the insets of panel b. The chromatographic

traces of the isolated decoy modules should be shown for comparison (panel b, top and middle chromatograms).

We see the point of the Reviewer and agree. We have followed the suggestion and included the chromatographic traces of the isolated decoy modules (Extended Data Fig. 2b, blue curves). For completeness, we also included corresponding SDS-PAGE strips to the right of the SEC profiles (after the ones for the peaks from the decoy module + FimH_L SEC runs).

Point 5

Panels c-g in Figure 1 cannot be understood without reference to Table S2. Information on the relevant disease mutations should be added to the figures or the legend.

We agree with the Reviewer and have added a paragraph to the legend of Fig. 1, explaining which UMOD mutations correspond to the GP2 residues shown in panels c-g. Please also note that a new panel c has been added that replaces the original panel f (see answer to point 1 by Reviewer 2).

Response to Reviewer #2:

Remarks to the Author:

The authors present an interesting study, inferring molecular details of filaments with an unusual bacterial decoy function. The overall approach is sound and I couldn't see any holes in the logic. It mixes cryo-EM, crystal, biochemical and bioinformatic approaches in a pleasing way, and includes a timely use of AlphaFold 2, both for Molecular Replacement and for cryoEM map fitting. The description is generally admirably clear and concise and the figures excellent. I have only a few comments

We are very grateful to the Reviewer for their appreciation of our work, and are pleased that they share our view that the combination of different experimental techniques with AF2 predictions can be truly powerful.

Point 1

Supp Table 2 focusing on pathogenic mutations is generally well-grounded (although we hardly need the structure to predict that disrupting (likely) disulfide bonds will be detrimental). A possible exception is P236L where the substitution does not change size and physicochemical characteristics dramatically. Is the Pro in an unusual area of the Ramachandran plot? Do methods like FoldX predict a significant loss of stability?

First, we agree with the Reviewer that the effect of the mutations affecting the Cys of UMOD D10C could have been guessed without a structure (although the latter revealed the disulfide bond connectivity of the domain, which was previously unknown); nonetheless, we thought it was important to mention them for completeness and, of course, because of their medical relevance.

Second, GP2 P126, which corresponds to UMOD P236, does not sit in an unusual area of the Ramachandran plot:

However, this invariant Pro and the His that immediately precedes it (conserved in both UMOD and GP2) play an important structural role in docking the long loop that connects β -strand D and E to the rest of D10C (in particular β -strand F and G). This can be seen in this figure, where GP2 loop residues G123-D130 are shown in cartoon representation and colored yellow whereas the rest of D10C is shown in both cartoon and surface representation and colored green/white:

In the right panel of the figure, GP2 residues that are identical in UMOD are underlined, whereas in the case of residues that are not identical the corresponding UMOD amino acid is reported in grey between parentheses.

As can be seen, the Pro in question inserts between a ridge formed by invariant V154 (V264 in UMOD) + T133 (V243 in UMOD), and invariant A156 (A266 in UMOD) + H162 (Y272 in UMOD) at the bottom, while invariant H125 inserts into a nearby pocket where it makes a hydrogen bond with invariant Y164. Simulating a Pro to Leu mutation on the graphics suggests that the substitution would cause clashes with either T133/V243 or D130/E240 (depending on the Leu rotamer), and thus disturb this part of the structure. Accordingly, FoldX analysis suggests that the GP2 P126L/UMOD P236L mutations would destabilize the structure of their respective D10C domains (predicted $\Delta\Delta G$ s 4.51 kcal/mol [± 0.18 kcal/mol] and 4.25 kcal/mol [± 0.02 kcal/mol]).

Although these considerations support the reported pathogenicity of UMOD P236L, we agree with the Reviewer that this was far from obvious from panel f of our original Fig. 1. In the revised version of the manuscript we therefore decided to only mention this particular mutation in Supplementary Table 2 (which has been expanded to cover additional mutations) and substitute

its panel (as well as the corresponding panel in Extended Data Fig. 5) with one showing D61 and P62, two other invariant residues whose corresponding amino acids in UMOD are affected by more immediately clear mutations (D172H and P173L, which have also been added to Supplementary Table 2). We think that these mutations are particularly interesting from a structural point of view, because they impact the interface between the β -hairpin and D10C by hitting residues that are at the very beginning of the latter.

Point 2

the homology modelling on p.17 has some issues. Although sequence identity is high it could have been done more carefully: different positions of the 1-residue deletion could have been trialed (the alignment does not seem completely unambiguous here); more advanced model quality measurements than DOPE could have been used; it is well known that standard MD (i.e. without constraints and using standard force fields) is more likely to degrade a model than refine it. However, since the results were only used as guidance I think these issues can be addressed simply by removing the word 'refinement' from 'refined by molecular dynamics in YASARA' and replacing it with a phrase indicating exploration of different conformations.

We understand the point of the Reviewer and, indeed, we only used the homology models for considering possible alternatives during rebuilding. We have therefore followed their suggestion and modified the sentence to “The respective models with the best DOPE scores⁴⁰ were then used as starting points for exploring different possible conformations by molecular dynamics in YASARA⁴¹.”

Point 3

especially given the excitement around AF2 it would be interesting to see somewhere a comment as to whether the structure would have been determinable if the crystal structure had yielded to experimental phasing, or if AF2 had an essential second participation in predicting inter-domain orientations when interpreting the cryoEM maps

As we mentioned in the main text (page 2, lines 38-40) and Methods (pages 15-16, section “Experimental phasing attempts”) of the manuscript, we tried quite extensively to solve the structure of the P1 crystal form by experimental phasing before being able to phase its data by MR with an AF2 model. Moreover, although this was not specifically pointed out in the original manuscript, the crystallization conditions of the C2 form contained four different heavy atoms (Er/Tb/Yb/Y), but none of these bound to the protein either, as shown by the lack of anomalous signal in the collected data. We have now made the latter clear, by adding an additional sentence at the end of the “Experimental phasing attempts” section: “*Similarly, no heavy atom bound to the C2 crystal form of the protein despite the fact that this was obtained in the presence of a mixture of different lanthanides and yttrium.*” (page 16, lines 260-262). Since the C2 and P2₁2₁2₁ crystal

forms of the protein diffracted to high resolution and provided much better quality data than the P1 form, we expect that ultimately we would have managed to solve at least one of them by S-SAD; however, both of these crystal forms were obtained at later stages of the project and, at that specific moment in time, we had no quick access to a tunable beamline. Thus, practically speaking, AF2 did make a very significant difference in terms of how fast we could phase our crystallographic data. Based on a growing number of reports, this very much seems to be the case for several other groups too.

It is also clear that having access to an AF2 model of the complete UMOD branch gave us a significant head start in terms of interpreting the low-resolution cryo-EM map of full-length UMOD (and, subsequently, that of the UMOD/FimH_L complex), even if at the time we had no access to PAE plot information about the reliability of the predicted inter-domain orientations. Nonetheless, as described in section “Cryo-EM map fitting, model refinement and validation” of the Methods, it was straightforward to flexibly fit the AF2 model of the UMOD branch into the cryo-EM map by using a tool like Namdinator.

In conclusion, we agree with the Reviewer that it is worthwhile adding a general comment about these aspects and have done so at the end of the main text: *“From a general point of view, this work provides an example of how deep learning techniques can significantly aid the X-ray crystallographic and cryo-EM investigation of challenging biological samples, by providing accurate models that can be used to solve the phase problem and aid the fitting of low resolution density maps, respectively.”*

Point 4

again given the AF2 excitement I would be interested to know

-- if the PAE predicted that the hairpin and the D10C domain relative orientation was not guaranteed to be correct

Included below is a plot of the PAE for GP2, generated using the AlphaFold Colab. The region between the red lines corresponds to the residues in the hairpin, while the large square in the lower right corresponds to the D10C domain.

In this instance it would be difficult to draw a definitive conclusion from the PAE plot. The PAE for aligning on a residue in the hairpin and scoring a residue in the D10C domain is generally higher than for residue pairs taken from within the D10C domain, but lower than for pairs involving the low-confidence N-terminus.

A potentially interesting observation is the asymmetry in the PAE for the hairpin. There is a higher PAE for aligning on residues in the hairpin and scoring residues from the domain than the other way around. One possible interpretation (supported by the comparison of the different crystal forms - see upper panel of Extended Data Fig. 4) is that the hairpin might be somewhat flexible, making it an unsuitable region to use for structure alignment.

-- if the Cys residues were modelled in such a way that software like PyMOL directly inferred disulphide bonds. As I understand it, disulphide bonds are not explicitly featured in AF2 outputs.

The AlphaFold neural network does not handle cysteines or disulfide bonds in any special way. However, since there are many examples of this feature in the PDB training data, the network has learned about disulfides and will often predict them reasonably well directly.

After the neural network prediction is generated, our pipeline performs constrained relaxation of the structure using OpenMM with an Amber99sb force field. The motivation is to remove any remaining clashes or unphysical bond lengths / angles. OpenMM does assign disulfides based on proximity, and this is the only part of the pipeline that is explicitly aware of them.

At the end of this full process disulfides in the predicted structure will usually be immediately apparent in PyMOL.

Point 5

minor point. In ED fig 4 upper panel, the blue domain linker between hairpin and D10C is practically invisible. Might a different superposition eg on D10C domain alone make it visible?

The β -hairpin/D10C domain linker of the AF2 model was indeed hardly visible in the original figure, due to the fact that it essentially overlaps with the corresponding part of the experimental structures. After exploring some alternatives, we decided to make the path of the AF2 model generally more visible, by increasing the transparency of the experimental structure models. This is a side-by-side comparison of the linker region before and after the change (with a red arrow indicating the linker itself):

Additional changes

Title

The title has been modified to “*Structure of the decoy module of human glycoprotein 2 and uromodulin and its interaction with bacterial adhesin FimH*” to make it clearer that the manuscript not only describes the structure of the decoy module, but also contains significant information about its interaction with FimH.

Fig. 2a

The format of the arrows in the panel has been made consistent.

Extended Data Fig. 1b

The sequence alignment panel has been modified so that it also specifies UMOD residue numbers (in grey, between parentheses) above the alignment itself, and the figure legend has been updated accordingly. We think that this small addition will make it easier for readers to locate residues for both proteins, rather than just GP2.

Methods/Structure solution by molecular replacement with AlphaFold2 models

Although in the original version of the manuscript we reported how we calculated GDT_TS scores, we forgot to include the actual scores themselves. This omission has now been fixed by adding a new sentence at the end of this section: “*Using these coordinates as a reference, the top ranked AlphaFold2 model had a Global Distance Test (GDT_TS) score of 94.9 (or 97.2 if only the D10C domain is considered).*”

Methods/Cryo-EM data collection

The sentence “*For collecting cryo-EM data from the UMOD-FimH_L complex (Supplementary Table 4),*” was modified to “*For collecting cryo-EM data from the UMOD-FimH_L complex (Supplementary Table 4), prepared as described in the section “Protein expression and purification,”* to make it clearer where in the manuscript is specified how the complex was prepared.

Methods/Cryo-EM map fitting, model refinement and validation

To fully comply with the data availability policy of the journal (“All data used in accepted papers should be available via a public data repository”), we have also deposited in EMDB/PDB the maps and model of the UMOD/FimH_L complex shown in Fig. 2d and Extended Data Fig. 8, and reported the corresponding accession numbers in the “Data availability” section of the manuscript. In connection with this, we have expanded the last paragraph of the “Cryo-EM map fitting, model refinement and validation” section of the Methods to provide more details about how we

assembled the UMOD/FimH_L complex model; added a new corresponding table (Supplementary Table 5); and submitted to the journal the corresponding full PDB validation report together with this manuscript revision.

Acknowledgements

Added “Akos Vegvari (Karolinska Institutet Proteomics Biomedicum core facility) for the MS analysis of the FimHL bands” and “the Plateforme d’Analyses des Glycoconjugués (PAGés) and the Plateforme d’Analyse Protéomique et de Protéines Modifiés (P3M) for GP2 N65 glycan LC-MS/MS”.

Supplementary Table 2

The table has been expanded to include all the D10C mutations for which we could find a reference; in addition, some of the descriptions of the predicted effect of the mutations have been made clearer.

Decision Letter, first revision:

9th Nov 2021

Dear Dr. Jovine,

Thank you for submitting your revised manuscript "Structure of the decoy module of human glycoprotein 2 and uromodulin and its interaction with bacterial adhesin FimH" (NSMB-BC45335A). It has now been seen by the original referees and their comments are copied below. The reviewers find that the revisions have fully addressed their prior concerns, and therefore we'll be happy in principle to publish it in Nature Structural & Molecular Biology, pending minor revisions to comply with our editorial and formatting guidelines.

To facilitate our work at this stage, we would appreciate if you could send us the main text as a Word file. Please make sure to copy the NSMB account (cc'ed above).

Thank you again for your interest in Nature Structural & Molecular Biology. Please do not hesitate to contact me if you have any questions.

With kind regards,

Beth

Beth Moorefield, Ph.D.
Senior Editor

Nature Structural & Molecular Biology

Reviewer #1 (Remarks to the Author):

The authors have carried out a very thorough revision and have successfully addressed all of my points. The manuscript should now be published as it is. The MS data on the GP2 mannose glycan are a significant addition to what was already an excellent piece of work.

Reviewer #2 (Remarks to the Author):

thanks for making these changes, no further suggestions

Author Rebuttal, first revision:

Response to the Reviewers

Reviewer #1:

Remarks to the Author:

The authors have carried out a very thorough revision and have successfully addressed all of my points. The manuscript should now be published as it is. The MS data on the GP2 mannose glycan are a significant addition to what was already an excellent piece of work.

Reviewer #2:

Remarks to the Author:

thanks for making these changes, no further suggestions

We are glad that both reviewers are satisfied with the changes that we introduced in the manuscript to address their comments. We would like to thank them again for their useful suggestions and a very constructive reviewing process.

Final Decision Letter:

21st Jan 2022

Dear Dr. Jovine,

We are now happy to accept your revised paper "Structure of the decoy module of human glycoprotein 2 and uromodulin and its interaction with bacterial adhesin FimH" for publication as a Brief Communication in Nature Structural & Molecular Biology.

As soon as your article is published, you can generate your shareable link by entering the DOI of your article here: http://authors.springernature.com/share.

Corresponding authors will also receive an automated email with the shareable link

Note the policy of the journal on data deposition:

<http://www.nature.com/authors/policies/availability.html>.

Your paper will be published online soon after we receive proof corrections and will appear in print in the next available issue. You can find out your date of online publication by contacting the production team shortly after sending your proof corrections. Content is published online weekly on Mondays and Thursdays, and the embargo is set at 16:00 London time (GMT)/11:00 am US Eastern time (EST) on the day of publication. Now is the time to inform your Public Relations or Press Office about your paper, as they might be interested in promoting its publication. This will allow them time to prepare an accurate and satisfactory press release. Include your manuscript tracking number (NSMB-BC45335B) and our journal name, which they will need when they contact our press office.

About one week before your paper is published online, we shall be distributing a press release to news organizations worldwide, which may very well include details of your work. We are happy for your institution or funding agency to prepare its own press release, but it must mention the embargo date and Nature Structural & Molecular Biology. If you or your Press Office have any enquiries in the meantime, please contact press@nature.com.

Please note that *Nature Structural & Molecular Biology* is a Transformative Journal (TJ). Authors may publish their research with us through the traditional subscription access route or make their paper immediately open access through payment of an article-processing charge (APC). Authors will not be required to make a final decision about access to their article until it has been accepted. [Find out more about Transformative Journals](https://www.springernature.com/gp/open-research/transformative-journals)

Authors may need to take specific actions to achieve compliance with funder and institutional open access mandates. For submissions from January 2021, if your research is supported by a funder that requires immediate open access (e.g. according to [Plan S principles](https://www.springernature.com/gp/open-research/plan-s-compliance)) then you should select the gold OA route, and we will direct you to the compliant route where possible. For authors selecting the subscription publication route our standard licensing terms will need to be accepted, including our [self-archiving policies](https://www.springernature.com/gp/open-research/policies/journal-policies). Those standard licensing terms will supersede any other terms that the author or any third party may assert apply to any version of the manuscript.

With kind regards,

Beth

Beth Moorefield, Ph.D.
Senior Editor
Nature Structural & Molecular Biology